# Marine plankton show threshold extinction response to Neogene climate change

Sarah Trubovitz [1✉], David Lazarus [2✉], Johan Renaudie [2] & Paula J. Noble [1]

Ongoing climate change is predicted to trigger major shifts in the geographic distribution of marine plankton species. However, it remains unclear whether species will successfully track optimal habitats to new regions, or face extinction. Here we show that one significant zooplankton group, the radiolaria, underwent a severe decline in high latitude species richness presaged by ecologic reorganization during the late Neogene, a time of amplified polar cooling. We find that the majority (71%) of affected species did not relocate to the warmer low latitudes, but went extinct. This indicates that some plankton species cannot track optimal temperatures on a global scale as assumed by ecologic models; instead, assemblages undergo restructuring and extinction once local environmental thresholds are exceeded. This pattern forewarns profound diversity loss of high latitude radiolaria in the near future, which may have cascading effects on the ocean food web and carbon cycle.

[1] Department of Geological Sciences & Engineering, University of Nevada—Reno, Reno, NV, USA. [2] Museum für Naturkunde, Leibniz-Institut für Evolutions- und Biodiversitätsforschung, Berlin, Germany. ✉email: strubovitz@nevada.unr.edu; david.lazarus@mfn.berlin

Numerous ecological models predict imminent shifts in the geographic distribution of marine plankton due to climate change (e.g., refs. [1–6]), but have not fully addressed the risk of global extinctions[7]. Global cooling over the last 10 million years (Ma)[8] would, according to these models, have caused polar plankton species to move equatorward to conserve their preferred habitat. Supporting this, paleontological work on marine sediments has found major changes in community structure and species richness decline (~35% of species) among high-latitude Polycystina (radiolaria)[9], a ubiquitous group of microzooplankton that are known to be important contributors to ocean ecosystems and geochemical cycles[10–12]. However, without a comparable low-latitude record, it has until now been impossible to determine whether high-latitude diversity decline was due to range contraction and migration versus permanent global extinction.

Radiolaria are marine microzooplankton (Rhizaria[13]) that comprise a significant component of ocean biomass[14], export carbon to the deep ocean[11,15,16], contribute to the global silicon cycle[17], and are among the most genetically diverse clades in the photic zone[12]. Despite recent advances in our understanding, radiolarians' climate sensitivity and species richness are poorly known[18]. The World Register of Marine Species[19] reports a total of 417 described living radiolarian species, and the global biogeography of 307 of these has been studied[20,21]. However, these values are likely significant underestimates, considering that molecular DNA analyses show ~1000 polycystine radiolarian taxa in ocean photic zones alone[12]. Whereas some of this genetic diversity may be accounted for by cryptic species, it is generally accepted that we have an exceptionally poor understanding of radiolarian species richness and ecology compared to related groups, such as the planktonic foraminifera[22]. Lacking reliable estimates of past and present radiolarian species richness, we have so far been unable to robustly determine how this vital plankton group responded to climate in the past, or to predict the impacts of anthropogenic change in the future.

Radiolarians, and many other holoplankton groups, have an unusually simple global biogeography. The vast majority of morphospecies are distributed throughout circumglobal latitudinal bands that approximate the first-order global planktonic biomes[20,23] (see "Methods"). Many radiolarian species (with the exception of some forms carrying photosynthesizing symbionts) do not require sunlight, and their distribution tends to depend, in addition to ocean water mass boundaries, on vertical food flux in surface to intermediate waters[24,25]. A single species may exist at multiple depths depending on latitude, in a phenomenon known as isothermal submersion[26,27]. As thermohaline circulation drives cool, subpolar-temperate water masses beneath warm, stratified waters of the tropics, some radiolarian species that occur near the surface at mid-high latitudes are known to follow isothermal pathways into intermediate and abyssal water depths at low latitudes (Fig. 1a). Isothermal submersion is believed to play a significant role in the composition of low and mid latitude radiolarian faunas, contributing 20–30% of species inventories[25]. Therefore, while radiolarian biogeography is generally partitioned into latitudinal provinces (endemic polar, mid-low latitude, and tropical) in the surface water (~0–200 m), there is a degree of lateral connectivity between them at depth[26]. Longitudinal gradients in species abundances are common, but true within-latitudinal biome endemicity is exceedingly rare (see "Methods"). Thus, data from a single section can in principle give a reasonable approximation of radiolarian species occurrences and richness over time throughout a given biome.

Fossil tropical radiolarian species richness for the late Neogene–Recent has never been fully surveyed[28]; to our knowledge, it has only been estimated from syntheses of the literature,

which is focused largely on common species and biostratigraphic marker taxa[22] (Supplementary Data 2). Our study utilizes an exhaustive approach to record (almost) all species preserved. Radiolarians are morphologically diverse, and include many rare and small species (~50 μm). These factors make it difficult to capture their full species richness using traditional enumeration and sieving protocols, which rely on fixed counts of a few hundred specimens of selected common species per sample (for studies of paleoenvironmental conditions derived from assemblage composition) and coarse (63 μm) mesh sieves[22]. However, it is critical that small or rare species be included in our tropical census, because they may represent refugial populations of polar species that shifted their ranges toward the equator during late Neogene cooling.

Here, we compare radiolarian diversity dynamics at low versus high latitudes to infer the evolutionary and ecological impacts of differential regional climate change over the last 10 million years. We construct a comprehensive species richness curve of low latitudes using tropical and isothermal submergence faunas from International Ocean Discovery Program (IODP) Site U1337 in the eastern equatorial Pacific (EEP), and examine it in conjunction with the only other fully censused Neogene to Recent radiolarian diversity history, from the high-latitude Southern Ocean (SO)[9]. Assemblage comparisons between the SO and EEP allow us to estimate how many of the species that disappeared from high latitudes ultimately survived global cooling by contracting or shifting their ranges to low latitudes, possibly via isothermal submergence pathways, or instead went globally extinct. Both high and low-latitude datasets are also compared to regional temperature proxy records[8,29,30] to evaluate radiolarian responses under different magnitudes of climate change. While the EEP experienced changes in temperature, productivity, weathering rates, and tectonic configuration[31] over the last 10 million years, the magnitude of these were relatively minor compared to the amplified climate change and variability experienced in the higher latitudes[8]. Our results show that SO assemblages underwent a severe decline in species richness (~35%), preceded by ecological restructuring, which we interpret as a threshold response to relatively high-magnitude temperature change. By contrast, species richness and ecological structure remained stable in the EEP. Furthermore, we find that the tropics did not serve as a habitat refuge for 71% species extirpated from the SO, indicating that marine plankton species cannot always track preferred habitats on a global scale and are instead acutely vulnerable to extinction during intervals of considerable climate change.

## Results & discussion

**Latitudinal comparison of richness and ecologic structure.** In order to generate a reasonably complete estimate of tropical radiolarian species richness over the last 10 million years, we made use of large sample sizes (~5000 specimens per sample) and completeness-based enumeration protocols[32–34] (see "Methods") to ensure adequate documentation of the full assemblage, including rare species. We also used finer mesh sieves (45 μm) than previous studies (e.g., refs. [35,36]) to make sure small species were retained. We determined species richness in two ways: within-sample and extrapolated. Within-sample richness is simply the number of species observed per sample, and when combined with standardized coverage, is the most appropriate metric for detecting patterns in species richness over time[32]. Within-sample richness was very uniform; mean = 356 species (±19) over the last 10 Ma (Fig. 2a). Extrapolation was used to estimate true species richness based on sample completeness[34]. In addition to estimating the number of rare species unobserved in each sample, this method allows for optimal species richness comparison

**a** Generalized ocean circulation & radiolarian biogeography

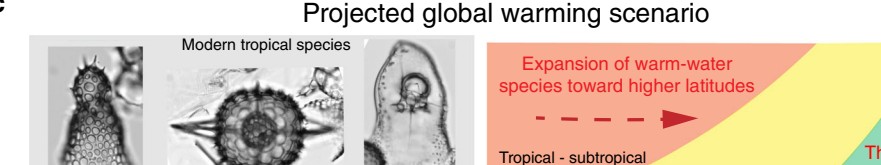

**b** Observed response to late Neogene cooling (<5 Ma)

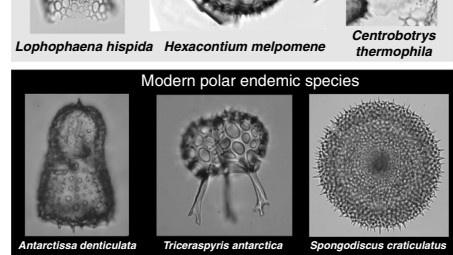

Tropical refuge species

*Tholospyris scaphipes*    *Lithomelissa mitra*    *Acrosphaera spinosa*

Polar extinctions

*Cycladophora spongothorax*    *Actinomma eldredgei*    *Lithomelissa stigi*

45 species found refuge in the tropics

119 species went extinct

Some species followed their preferred habitat toward the tropics as global temperature dropped

**c** Projected global warming scenario

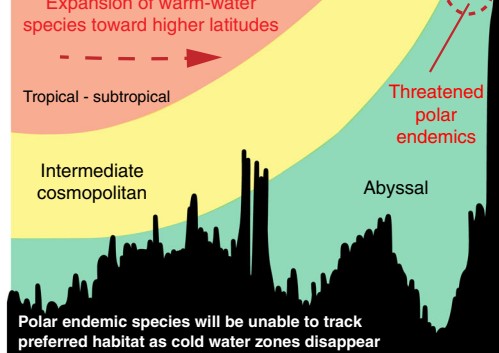

Modern tropical species

*Lophophaena hispida*    *Hexacontium melpomene*    *Centrobotrys thermophila*

Modern polar endemic species

*Antarctissa denticulata*    *Triceraspyris antarctica*    *Spongodiscus craticulatus*

Expansion of warm-water species toward higher latitudes

Threatened polar endemics

Polar endemic species will be unable to track preferred habitat as cold water zones disappear

among samples that vary in community structure or sample size[34]. Extrapolated species richness had a mean of 485 (±36) over the last 10 Ma (Fig. 2a). For comparison to previous work, we computed 10–0 Ma within-sample species richness for the tropical Pacific region (all longitudes, ≤20°N/S, all sites labeled Pacific for ocean) from the NSB database of the published literature[37]. This dataset includes radiolarian species occurrences in 1541 samples from 26 sites in the tropical Pacific, making it a

**Fig. 1 Radiolarian biogeography with observed and predicted responses to temperature change. a** Illustrates generalized radiolarian provinces[20, 26] and their relationship to water mass temperature (warm versus cool color shading) and circulation (gray arrows). Due to high-latitude water mass submergence under warm, stratified waters in lower latitudes, radiolarian species occupy habitats at multiple latitudes, and depths throughout the world oceans. Thus, marine sediments from the tropics reflect a composite of several vertically stacked faunal assemblages, some of which are contiguous with higher latitude surface assemblages. Sediments beneath polar waters include cosmopolitan deep-water radiolarians, as well as high-latitude endemic surface water species. Stars in (**a**) indicate the latitudes sampled for this study, and the gray bars highlight the radiolarian assemblages included in each sedimentary composite. The horizontal purple bars indicate latitudes known for good radiolarian (silica) preservation, based on surface sediment composition[22]. Our data show that some species were extirpated from high latitudes but persisted in the tropics during the late Neogene, either by migration or range restriction (**b**). With predicted global warming, modern Southern Ocean species will not be able to use migration or range contraction to escape environmental stressors, because their preferred cold-water habitats are disappearing from the globe (**c**). However, tropical endemic species may expand their ranges toward midlatitudes. The color polygons in all three panels represent generalized radiolarian biogeographic provinces, as well as their relative water mass temperatures (cooler colors indicate cooler temperatures, and vice versa). Globe image adapted from NASA Blue Marble: Next Generation imagery. Ocean floor bathymetry from Google Earth™ seafloor elevation profile (5°N–74°S, at 120°W).

robust estimate of overall trends for those species reported in the literature, even if samples were not fully surveyed for richness. Analysis of NSB data showed stable richness levels (63 species ± 10, per 1 million-year time bin) across the study interval, albeit at significantly lower species richness than our survey (Fig. 2a). Together these results indicate high, near constant, tropical species richness over the last 10 million years, with extrapolated richness nearly five times greater than previously reported (Supplementary Data 2).

In addition to species richness, the relative abundances of species within tropical paleocommunities remained stable, as measured by the Pielou equitability index[38] (mean = 0.84 ± 0.02) (Supplementary Data 1). This index measures the relative proportion of taxa in a community on a scale of 0–1, with 0 being entirely dominated by one taxon, and 1 being a maximally equitable distribution of individuals among the taxa. From these results it is clear that both species richness and ecological structure have remained stable in the tropics despite modest temperature decrease[31] (Fig. 2a).

The SO exhibited significantly lower radiolarian species richness than the tropics over the last 10 million years, consistent with the global latitudinal biodiversity pattern observed for most taxa[39], and for modern radiolarians[21]. In contrast to tropical diversity stability, SO assemblages underwent a severe decline in species richness over time[9] (Fig. 2b). The loss of species richness in the SO was presaged by the rising ecological dominance of the genus *Antarctissa*, which comprised <20% of SO radiolarian communities around 10 Ma, but nearly tripled in abundance to >40% by 7 Ma[9], without any significant change in species richness (probability of no change in species richness versus time during 8.5–5 Ma: $p = 0.92$ [raw], and $p = 0.36$ [extrapolated], both two-tailed $t$-tests with 9 degrees of freedom). This shift in community structure is captured by the Pielou equitability index, which dropped by ~0.1 in the span of only 1 million years (between 9 and 8 Ma; Fig. 2b), and remained low throughout the rest of the Miocene and Pliocene. Many of the species that became rare as *Antarctissa* became common disappeared completely from the assemblage during the 3–4 million years that followed (Fig. 2b), producing a significant decline in richness (probability of no change in species richness versus time during 5–0 Ma: $p = 8.55e − 10$ [raw], and $p = 3.30e − 5$ [extrapolated], both two-tailed $t$-tests with 42 degrees of freedom). Changes in community structure such as increasing dominance of a few species may be a precursor to extinction, and have been reported to precede mass extinction events in the fossil record[40].

**Fate of extirpated SO species.** A closer look at the polar biodiversity loss through a cross-comparison with low-latitude species occurrences reveals that 71% of the 233 lost polar species went extinct. These species were not observed in coeval or younger low-latitude sediments, as would be expected if they had survived by tracking their preferred temperature conditions (Supplementary Data 3; "Methods"). The unobserved SO species were also absent from all other tropical Pacific sites (<20°N/S, all longitudes) in the NSB Database, with the exception of two early and middle Miocene species that were likely reworked specimens[22]: *Dorcadospyris alata* (Riedel, 1959) and *Didymocyrtis violina* (Haeckel, 1887). To determine the specific fate of each SO species, we found the geologically oldest and youngest occurrence of each species observed in both the SO and EEP, and marked it as present throughout this temporal range (the range-through method is commonly used in paleontology[41]). Unlike extrapolation or within-bin richness, the range-through method accounts for the taxonomic identity of all species that were likely present during a given time interval, even if they were too rare to be observed in some samples during data collection. The drawback of the range-through method is that it underestimates species richness and overestimates origination/extinction at either end of the time series, because species cannot be ranged through the oldest and youngest samples[42]. Therefore, range-through was only used when appropriate for geographic comparison of species presence–absence, whereas extrapolation was used to estimate species richness trends through time because it lacks the artifactual edge effect (see "Methods").

Cumulatively, 166 SO species went extinct from the Late Miocene (10 Ma) to Recent, with a net loss of 144 species due to offset from the origination of new, cold-adapted species. While 71% of the species suffered extinction, the remaining 29% of polar species loss is explained by extirpation from the SO (Fig. 1b). Extirpated species were either endemic and migrated to lower latitudes, or cosmopolitan and contracted their ranges to lower latitudes (outside the SO) (Fig. 3). SO extinction rates peaked between 5 and 4 Ma, but continued to be higher than the Miocene into the Late Pliocene and Pleistocene (Figs. 3 and 2b). Low-latitude range contractions were also highest between 5 and 4 Ma, but the number of species that took refuge in low latitudes was substantially less than the number of extinctions. In total, less than 1/3 of the SO species impacted by high-latitude cooling (67 species) successfully took refuge in the deep tropical oceans following their extirpation from high latitudes. This is presumably because polar species that depend on surface or intermediate water conditions (e.g., food availability or sunlight for photosynthetic symbionts) cannot survive deep submergence in the tropics. Such a phenomenon was observed in the euphausiid species *Nematoscelis megalops*; Northwestern Atlantic populations that followed cold-water parcels as they submerged and ultimately died out due to lack of suitable food at depth[43]. We found that only a small fraction of radiolarian species truly migrated from endemic distribution at high latitudes, to endemic

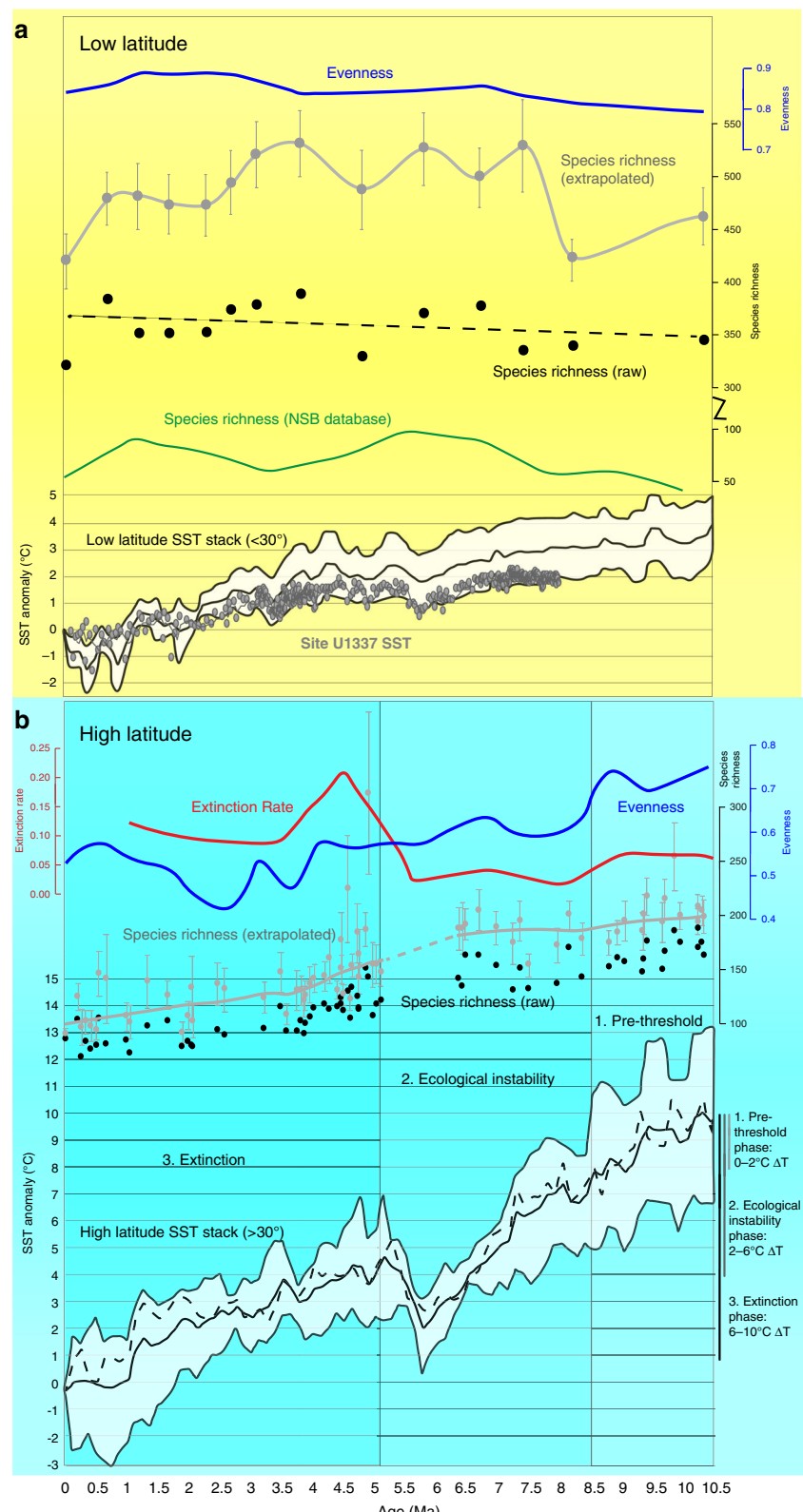

distribution at low latitudes (<1%, 12 species out of 233; Supplementary Data 3), contrary to what modern ecological models would predict. Most models parameterize biogeographic response primarily by changing temperature, the most significant environmental predictor of radiolarian and other plankton species distributions, thus predicting gradual northward shifts toward the tropics in response to global cooling for SO species

(e.g., refs. [3,5,6,44]). However, our observed response pattern shows that these ecologic range migration models may not realistically scale up to evolutionary processes over long time periods. Our findings suggest that SO radiolarian species are unable to consistently track their preferred temperature conditions, possibly due to biotic factors or surface water hydrographic barriers, such as the polar front.

**Fig. 2 Radiolarian evolution and sea surface temperature (SST) over the last 10 million years.** In both panels, black points indicate raw species richness observed in each sample. Extrapolated species richness (asymptotic diversity estimates) are shown in gray. Error bars indicate the mean (center) ± standard error, based on 500 bootstrap replicates. Extrapolation was performed using the R package iNext[66] (Hill numbers of order $q = 0$), and is based on sample completeness[34]. A lowess smoother (span = 0.33) is applied to the lower panel to aid with visualization; a dashed line is used across the data gap. Evenness (Pielou equitability index[38]) is plotted in blue, and boundary-crosser extinction rate[70] is plotted in red (most recent time bin removed due to edge effect, see "Methods"). The green line in (**a**) refers to tropical Pacific radiolarian diversity recorded in the Neptune Sandbox Berlin (NSB) database[37], which includes data from previous deep-sea drilling research (note broken scale). Southern Ocean radiolarian data were obtained from[9] and re-analyzed for this study using updated age models in NSB. Temperature data are re-plotted from[8, 29], and are based on alkenone biomarker proxy reconstructions. In (**a**) the SST anomaly curve indicates mean tropical SST over time and one standard deviation from the mean, as reported in[8]. The gray data points show SST data from Site U1337 only[29]. In (**b**), the black line indicates the mean SST for compiled northern hemisphere and southern hemisphere mid and high latitudes[8]. The dashed line indicates the median value among these datasets, and the envelope refers to the maximum and minimum average temperatures among these datasets. Due to some uncertainty in Southern Ocean age models, only broad patterns on the scale of ~1 million year (Ma) should be considered in (**b**), rather than precise temporal correlations between radiolarian evolution and temperature. The gap in sampling between ~5 and 6.5 Ma shown in (**b**) is due to a regional sedimentation hiatus and high age model uncertainty of this specific time interval at the Southern Ocean sites. Different shades of blue background in (**b**) refer to phases of Southern Ocean radiolarian response (see text).

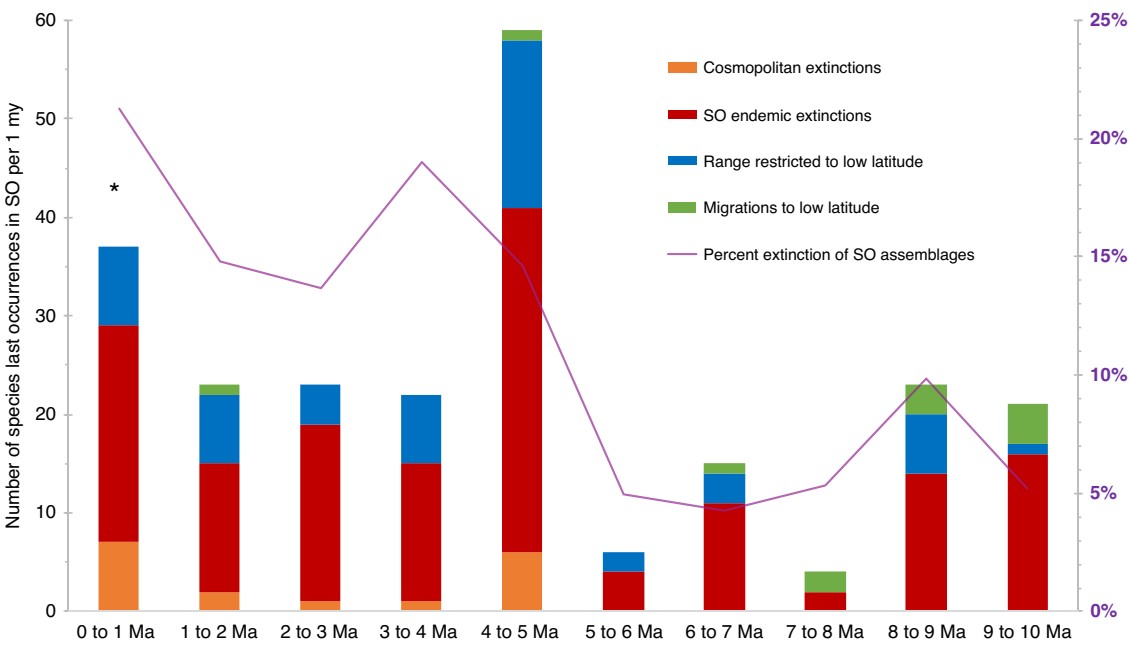

\* Extinctions from 0 to 1 Ma are likely overestimated due to range through edge effect

**Fig. 3 Fate of species that disappeared from Southern Ocean assemblages during the late Neogene.** Last occurrence datums of Southern Ocean (SO) species were compiled into 1 million-year bins, after each species was ranged through its first and last occurrence in the dataset (see "Methods"). Last occurrences were cross-checked against the low-latitude range-through species dataset, to determine whether the SO species were extirpated to warmer climates or went globally extinct. Warm colors represent extinctions and cool colors represent extirpations. Red bars = number of endemic SO species that became globally extinct; orange bars = cosmopolitan species that went extinct; blue bars = cosmopolitan species that restricted their range to the low latitudes; and green bars = endemic SO species that relocated from the SO to the low latitudes (migrations). The purple line indicates the percentage of the SO species assemblage that suffered extinction in each time bin (species that did not persist to the next time bin in the SO, nor were extirpated to low latitudes via migration or range restriction; see "Methods"). Overall, extinction was the main cause of SO diversity decline throughout the study interval. Migration was the least common outcome for species that disappeared from the SO. The highest extinction rate occurred between 5 and 4 million years (Ma), but high extinction continued into the Late Pliocene and Pleistocene. From 0 to 1 Ma, the number and percentage of extinctions may be an overestimate, because it is not possible to range-through species in the most recent time bin (see "Methods"). Therefore, some species may appear to have gone extinct, but were actually just too rare to be observed in the most recent samples. The data used in this figure are reported in Supplementary Data 3. A list of low-latitude species from the eastern equatorial Pacific is given in Supplementary Data 4.

**Threshold response to temperature change.** We attribute the late Neogene SO extinction interval (<5 Ma) to relatively high-magnitude polar climate change, compared to the tropics[8] (Fig. 2). Although there is no direct SST proxy record from the SO, a compilation of mid and high-latitude temperature data from both hemispheres indicate an average of 10 °C total polar SST cooling over the last 10 Ma[8], with several shifts of ~3 °C over durations of 2–3 million years (Fig. 2b). Plankton in the SO were exposed to much greater temperature change and oscillation than those in the tropics. The gradual change of ~3 °C in the EEP, as indicated by paleotemperature proxies[8,29], appears to be below the extinction sensitivity threshold of tropical radiolarians, whereas a relatively large inferred drop of ~10 °C with high intermittent variability over the last 10 Ma in the SO elicited a major extinction response once temperature thresholds were surpassed. Figure 2b illustrates the SO radiolarian temperature response, which we separate into three phases, based on changes in community evenness and extinction rates. In Phase 1 (10.5–8.5

Ma) the SO underwent an inferred 2 °C drop in SST, with no apparent effect on radiolarian communities. Phase 2 (8.5–5 Ma) is defined by ecological instability with a stark decline in community evenness, as temperature dropped by an additional 4 °C. This phase marks the beginning of the SO radiolarian temperature response; the increased rarity of many species likely made these communities vulnerable to later biodiversity loss[40]. In Phase 3 (<5 Ma), substantial diversity decline (loss of 119 species) and elevated extinction rates occurred as inferred temperatures went from 6 °C to eventually 10 °C below Phase 1 levels. The mean extinction rate <5 Ma was 0.125 (excluding youngest edge bin), whereas the mean extinction rate of earlier Neogene time bins (22–5 Ma) was 0.032. SO extinction rates during Phase 3 were significantly higher than at any other time during the Neogene (Wald test; $F$ value: 24.4356, $p$ value: 0.0002; see "Methods"). This three-phase response to temperature suggests that polar radiolarians have an SST change tolerance threshold of ~6 °C, which, when exceeded, can lead to long-lasting ecological instability and extinction.

**Implications for modern oceans**. The Intergovernmental Panel on Climate Change predicts that global temperature rise due to anthropogenic activity will be especially amplified in polar regions, with severe impacts on high-latitude climate and ecosystems[45]. Modeling simulations indicate that SO SSTs are likely to increase by 7–10 °C in the span of only 300 years (1990–2290)[46], which is approximately the same magnitude of SST decrease incurred at high latitudes over the last 10 Ma[8]. This suggests that the cold-adapted radiolarians, diatoms[7], and other high-latitude plankton groups, will be at elevated extinction risk due to the relatively high magnitude and rate of anthropogenic climate change. Moreover, the directional trend of contemporary change (warming versus Neogene cooling) presents an additional challenge for high-latitude species, as it limits their potential to survive via extirpation. Modern cold-adapted radiolarians will not be able to reverse their late Neogene range-restriction and migration pathways to follow cold habitats, since these habitats are disappearing from the globe. Furthermore, the rapid rate of change limits the possibility of significant adaptation or evolutionary replacement of lost species. This combination of factors will likely make future extinction more severe than that of the late Neogene, which occurred slowly enough (over 5 million years) for origination of new species to partially offset those lost by extinction. The sudden loss of substantial numbers of cold-adapted radiolarian species could threaten biodiversity and ecosystem functioning in the postwarming SO for several million years to come[47]. Their absence would possibly diminish the efficiency of carbon export to the deep ocean[11,16], possibly contributing to weakened $CO_2$ draw-down from the atmosphere and a positive climate warming feedback. The loss of these species could also impact both primary producer communities and higher trophic groups, with unknown cascading consequences on the ocean food web. Unlike those in high latitudes, tropical endemic radiolarian species richness was not substantially affected by the relatively lower magnitude of temperature change at the equator. They may even temporarily expand their ranges toward the midlatitudes for the duration of the warming event (Fig. 1c), as suggested by patterns of other marine groups (e.g. refs. [48,49]).

The findings of this study are significant in that they suggest marine plankton temperature thresholds (~6 °C for radiolarians) may be surpassed within the next 300 years, with important consequences for species richness and ecosystem functioning at high latitudes. Our results document a previously un-recognized major radiolarian extinction interval over the last 5 million years in the SO, only slightly lower in magnitude to biodiversity decreases seen globally in some marine plankton groups (e.g., calcareous nannofossils, planktonic foraminifera) during the End Cretaceous mass extinction[50]. Living SO radiolarian assemblages have adapted to present cold conditions, following a long-term cooling trend over the last 10 Ma. With ongoing anthropogenic warming, these assemblages may face acute habitat loss and no escape route to colder regions. Rates of climate change will also be too rapid for sufficient evolutionary adaptation or origination to successfully offset extinction[51]. Given these compounding factors, we suggest that in the next several hundred years radiolarians could suffer an extinction similar or greater in severity to the one we have identified during the late Neogene. Our data support the idea that changes in ecological structure may be a precursor to future extinction events. Therefore, with close monitoring of high-latitude plankton, it may be possible to detect ecological instability before the hypothetical extinction threshold is breached. The information extractable from long-term records of species evenness and richness can provide an important context for evaluating modern ecosystem health and impending threat from climate change; we urge that such records be more commonly integrated with modern climate impact studies in the future.

## Methods

**EEP site selection & samples**. All Site U1337 samples were obtained from the IODP Gulf Coast Core Repository. Sample ages were linearly interpolated from an astronomically tuned high-resolution age-depth model[31] (Supplementary Fig. 1). In total, 14 sediment samples were analyzed for this study, spaced at ~0.5 and 1 million-year intervals throughout the last 10 Ma (Supplementary Data 1). Site U1337 was chosen for its consistent paleo-location near the equator over the last 10 Ma, between the North and South Equatorial Currents, and outside of the main equatorial Pacific Cold Tongue zone of high upwelling[29] (Supplementary Fig. 1). Due to the low magnitude of paleoceanographic change reconstructed at this site[29,31], it provides an ideal record of plankton in the absence of significant environmental change.

**Biogeographic rationale**. Our claim to have examined the occurrences of radiolarian species in the late Neogene tropics using data from a single geographic location in the Pacific depends on this location having a species inventory representative of tropical regions during this time interval. We believe this to be a reasonable assumption, based on what is known about biogeographic patterns in radiolarians, and in plankton more generally. As noted in the main text and documented here, the vast majority of species show circumglobal distributions within restricted latitudinal bands, so that within any latitudinal band (oceanic biome), all species from that biome can be expected to be present, albeit at varying degrees of abundance. This remarkably simple biogeography of marine holoplankton is due to the unique nature of the environment it lives in—a globally circulating fluid, which is well mixed on time scales of decades to centuries by the global wind driven circulation at the surface, and the thermohaline circulation at depth[22,52,53].

Radiolarian species distributions have been extensively documented in both the Neogene fossil record[22,37,54], and in the plankton over many decades[20]. Studies of fossil species have mostly concentrated on a fraction of the species richness—biostratigraphic marker species, and relatively common forms as tracers of ocean environments[22]. These studies, with the exception of a few species endemic to upwelling zones[55] have not reported longitudinal endemism within the tropics or midlatitudes. This negative observation must be tempered however by the incompleteness of the data, inconsistency of taxonomic name usage, and the somewhat limited geographic coverage by ocean drilling, particularly in older Neogene sediments[22]. Studies of surface sediment assemblages, representing the last few hundred years of plankton history, and of living radiolarians in the water column are geographically much more complete. These studies also have not noted any significant degree of longitudinal endemism. The recent monographic-scale synthesis of Boltovskoy et al.[20], which examined >300 species and nearly 7000 samples, and specifically tried to harmonize taxonomic usage, note "None of the species covered was scarce in the Pacific but abundant in either the Atlantic or the Indian oceans". They also reported (op. cit., Table 2[20]) only nine species (including subspecies) that were dramatically rarer in the Atlantic than the Indo-Pacific, and only three species (all missing in the Indian Ocean) that appear not to be present in all three ocean basins. By contrast, virtually all species (op. cit., Table 4[20]) showed at least some degree of differentiation with latitude. The biogeographic pattern of radiolarians is also mirrored in many other plankton groups[23,56,57], and in particular in the planktonic foraminifera, which are closely related to the radiolaria, and are undoubtedly the most exhaustively studied of all fossil-forming plankton clades. Among the ca. 50 species of living planktonic

foraminifera, only 4 species are endemic: three to the Indo-Pacific, and just one species (or morphologic subspecies: *Globigerinoides ruber* var. pink) to the Atlantic[58,59].

Lastly, we consider the difference between morphospecies units, as used in our study, and (possibly cryptic) genetic species. In recent decades, many holoplanktonic morphologically defined protist species have been shown to consist of more than one genetic species, particularly in planktonic foraminifera[59,60]. Radiolarian studies of intraspecies variation are very rare[61,62] and the extent of cryptic speciation remains unknown, but must be assumed to be at least occasionally present. Our study of species extinction in the fossil record, necessarily being based on morphospecies, is thus biased against finding extinction. Morphospecies listed as present in both low and high latitudes could differ at the genetic level and thus are not necessarily records of species survival. We can only record the loss of a morphospecies when all the genetic species that it may have contained have gone extinct. Our estimates of species extinction risk in plankton should thus be seen as a lower boundary, and the true level of extinction at the genetic level may well be substantially greater than we report.

**Slide preparation.** Slides were prepared in the Micropaleontology Lab at the Museum of Natural History in Berlin. Samples were soaked in a weak solution of a dispersant (sodium hexaphosphate), Hydrogen peroxide ($H_2O_2$), and 10% HCl were used to dissolve out the carbonate content. The remaining siliceous material was sieved at 45 μm in order to retain small species. A gelatin solution was applied to glass slide cover slips and allowed to dry, before being submerged in beakers of distilled water. A measured fraction of the rinsed >45 μm cleaned sample fraction was pipetted into beakers for random settling onto glass cover slips. After settling (~2 h), the distilled water was siphoned to near the slide surface, and the rest was evaporated out of the beakers using heat lamps. Once fully dry, the cover slips with settled material were attached to glass slides with warm Canada balsam. Finished slides were heated at 60 °C until dry enough to handle (~48 h). These procedures follow those of Renaudie and Lazarus[9], an adaptation of that originally developed by Moore[63]. All samples yielded very well to moderately well preserved radiolarian assemblages, as previously reported for this site[64]. Slides were observed under transmitted light at ×100–400 magnification. All counts were performed using an Olympus BX51 microscope. An OMax 10.0 megapixel mounted camera (model A35100U) with ToupLite imaging software was used for taking photographs.

**Taxonomy.** Over 86,000 images were taken of representative specimens of >1000 taxa (945 species-level groups), and organized into a digital catalog using Graphic Converter image browser software. This helped establish consistent taxonomic concepts throughout the data collection process, which was critical given the number of undescribed species encountered. More than half of the 945 species-level taxa observed in this study (540) are new species awaiting formal description (Supplementary Data 4). This taxonomic image catalog was regularly checked during counting to ensure consistency with taxonomic concepts used in the SO study[9]. The large majority of the categories are resolved to species level, but genus, subfamily, and family level categories were used to record incomplete or otherwise not species-identifiable specimens as precisely as possible. These higher-level taxonomic counts were not used in data analysis for this study, but are reported in Supplementary Data 1. Images of all described and undescribed species-level taxa are documented in Supplementary Data 7.

**Enumeration.** Raritas[65] software was used to record enumerations for each sample, and to observe collection curves during counting. After 2000 specimens had been counted, Raritas was operated in rare count mode. This setting automatically adds counts of common species (in our study we chose those that comprise >5% of the assemblage) based on their proportions in the initial count of 2000 specimens. Due to the evenness of EEP radiolarian assemblages, rare count mode only applied to ~3–4 counting groups per sample, none of them at the species level. Therefore, the vast majority of counts, including all used in data analysis for this study, were recorded manually.

Each sample was counted to ~5000 specimens, once visual inspection of the collection curve indicated that it had sufficiently flattened (apparent slope <30°, see Supplementary Fig. 2). This initial observation made during data collection was confirmed by calculating sample coverage for species-level data. Uniform and sufficient coverage was determined using the iNext R package[66] algorithm based on Good's U estimator[33,34], which indicated all samples had between 93% and 97% coverage. Exact coverage values for each EEP sample are reported in Supplementary Data 1. Our methods and coverage values are similar to those reported by Renaudie and Lazarus[9], however, coverage was slightly higher for the SO samples (~99%). Exact coverage values for SO samples used in this study are reported in Supplementary Data 5. Since more samples were counted from the SO than the EEP (69 and 14, respectively), there were overall more observations from the SO than EEP (500,309 and 73,290 specimens, respectively). Therefore, it is unlikely that incomplete sample coverage significantly contributed to the number of extinctions we observed in the SO.

**Biodiversity and ecology metrics.** Biodiversity and ecological metrics were calculated using species-level data only. Raw and range-through species richness were

computed using the R package divDyn[67]. Raw diversity, range-through diversity, extrapolated diversity, and evenness are reported for the EEP dataset in Supplementary Data 1, and for the SO dataset in Supplementary Data 5. Raw diversity is the number of species observed after ~5000 specimens were counted, the collection curve had flattened, and coverage was >90%. Range-through diversity differs in that it accounts for all species assumed to be present based on their stratigraphic ranges, even if they were not physically observed in every sample. The range-through algorithm in divDyn marks taxon presence in all time intervals between its first and last occurrence, thereby counting it as present regardless of whether or not it was documented in all intermediate time intervals. This practice is commonly used in paleontology[41], and was also used by Renaudie and Lazarus[9] in their SO radiolarian diversity census. Although this method is considered to be a good estimate of true biodiversity, it has significant edge effects on the oldest and youngest few samples[42]. Taxa cannot be ranged through these samples, as no, or insufficient data exists on either side, artificially causing diversity estimates to appear low for the oldest and youngest few samples in the time series. Therefore, we rely on other methods (raw within-sample diversity and extrapolation) to illustrate species richness trends through time (Fig. 2). Our use of range-through diversity was limited to extinction analyses (see "Methods" sections below), because it is the only technique capable of tracking the identity of individual species through time. All range-through diversity values, including those subject to analytical artifacts (marked with asterisks), are reported in Supplementary Data 1 (EEP) and Supplementary Data 5 (SO).

Extrapolation was used to estimate total species richness for each EEP and SO sample, had it been counted to completion (extended to asymptote of coverage = 100%). Because this computation accounts for sample coverage, it enables comparison of species richness between samples, even if they contain different numbers of specimens or display variable community structure[34]. Unlike range-through diversity, extrapolated diversity is not subject to edge effects on the oldest and youngest samples. However, there is significant statistical uncertainty inherent in each extrapolation estimate (reported as standard error). Coverage-based species richness extrapolation was performed in the R package iNext[66], using the parameters $q = 0$, and bootstrap replicates = 500. The results of this computation are listed in Supplementary Data 1 (EEP) and Supplementary Data 5 (SO), and are illustrated in Fig. 2.

The Pielou equitability index ($J$)[38] was used as a metric of ecological structure and evenness. This measures the degree of population distribution among taxonomic units on a scale of 0 (least even) to 1 (most even). Pielou equitability was calculated using the Paleontological Statistics Software Package (PAST) version 4.03[68]. Pielou equitability values are reported in Supplementary Data 1 (EEP) and Supplementary Data 5 (SO), and are illustrated in Fig. 2.

$$J = H/\log(S),$$

where $H$ = Shannon–Wiener diversity index[69]; $S$ = number of species in the sample.

**Southern Ocean extinction rate calculation.** SO species occurrence data were obtained from Renaudie[9], and vetted for high age model quality. Extinction rate was calculated based on Foote's boundary-crosser approach[70], which has been demonstrated to be a reliable method for paleontological occurrence data[71]. The mean extinction rate <5 Ma was 0.125 (not including youngest edge bin), whereas the mean extinction rate of earlier Neogene time bins (22–5 Ma) was 0.032. In order to test the statistical significance of the increase in the SO radiolarian extinction rate <5 Ma, we built a first-order autoregressive time-series model using generalized least squares[72]. Membership of the time bins was defined as either belonging, or not belonging, to the increased extinction rate timespan (<5 Ma), as a categorical independent variable. Significance of differences between timespans was determined using a Wald test[73], implemented using the function anova.gls from R package nlme[72]. The results (F value: 24.43556, p value: 0.0002) showed that the extinction rate values after 5 Ma were significantly different than the ones before (22–5 Ma).

**Extinction versus extirpation calculation.** This analysis utilized range-through datasets generated from EEP and SO occurrences in each 1 million-year time bin spanning 10–0 Ma (see "Biodiversity and ecology metrics" section above). When applicable, the last occurrence datum (LAD) of each SO species (list in Supplementary Data 6) was determined, for comparison to EEP species ranges (Supplementary Data 3). Each taxon that disappeared from the SO was cross-checked against EEP taxonomic occurrences (Supplementary Data 4) to determine whether it was ever present in the tropics, and if so, at what time. There were four measurable outcomes for each species with a LAD in the SO (results illustrated in Fig. 3): (1) it was not present in the EEP at any point during our study interval, in which case it was interpreted to be an endemic polar species that went globally extinct. (2) It was present in the EEP prior to disappearance from the SO, but not afterwards, in which was it was interpreted to be a cosmopolitan species that went globally extinct. (3) It was present in the EEP after it disappeared from the SO, but not before, indicating a true migration event. (4) It was present in the EEP both before and after its disappearance from the SO, suggesting it was once a cosmopolitan/deep-dwelling species that later restricted its range to the tropics. To ensure that taxonomic concepts were consistent among authors and regions, photographs

of specimens from both regions were closely examined to confirm that they were conspecific.

**Reporting summary**. Further information on research design is available in the Nature Research Reporting Summary linked to this article.

## Data availability

All data generated or analyzed during this study are available in the Zenodo repository (https://doi.org/10.5281/zenodo.4014322).

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

## Acknowledgements

We would like to thank the International Ocean Discovery Program (IODP) for sediment samples; U.S. Science Support Program (Schlanger Fellowship), Geological Society of America, Paleontological Society, Cushman Foundation, German Academic Exchange Service (DAAD), and Make Our Planet Great Again—German Research Initiative (MOPGRA-GRI) for funding; Fr. Sylvia Salzmann (Museum für Naturkunde) for sample preparation.

## Author contributions

S.T., D.L., and P.N. conceived and designed project and cowrote the paper. S.T. collected EEP radiolarian data and analyzed EEP and SO datasets. J.R. and D.L. provided SO radiolarian biodiversity dataset with updated age models from NSB, and assisted in taxonomic identifications and cross-check of EEP and SO species. J.R. performed SO extinction rate calculations and statistical test.

## Competing interests

The authors declare no competing interests.
