## [Peer Review File · Nature Communications]

Reviewers' Comments:

Reviewer #1:

Remarks to the Author:

Review of "Marine plankton show threshold extinction response to Neogene climate change" by Trubovitz, S., Lazarus, D., Renaudie, J., & Noble, P.J.

Based on analyses of radiolarian species richness changes throughout the last 10 million years at the equator and in the Southern Ocean, the authors conclude that a large proportion of the high latitude species have been unable to cope with global cooling by shifting their distributional ranges toward lower latitudes, and went extinct. Extrapolating these findings to the current global warming trend, they suggest that many living warm and temperate-water radiolarians may not be able to shift their ranges polewards or move to deeper (and cooler) layers in order to persist in their preferred thermal regime, but will also go extinct. Further, in a scenario of global warming, polar species will lack a refuge altogether, and will also conceivably disappear.

The work is definitely interesting, not only for the radiolarian community, but also for a much wider audience of biological and paleontological oceanographers, as well as for scholars interested in the effects of global warming on the biota. The materials and methods used are generally solid, and the conclusions are convincing.

I have only one major observation, and several minor suggestions.

The major point is the scarcity of datapoints for the equatorial site (13 samples, one per 0.5 Ma), which contrasts sharply with the wealth of information for the Southern Ocean. Given the exhaustive taxonomic work performed on the data, this difference probably does not invalidate the conclusions arrived at, but its potential implications deserve further justification. BTW, I would have expected to see, in the Supplementary tables, the two complete databases with presence/absence data for each species in the EEP and in the SO.

Several minor points follow.

"...contribute to the global silicon cycle..."

A more updated and more specific reference for this statement: Llopis Monferrer N, Boltovskoy D, Méndez Sandin M, Tréguer P, Not F, Leynaert A (2020) Estimating biogenic silica production of Polycystina and Phaeodaria (Rhizaria) in the global ocean. *Global Biogeochemical Cycles*, 34:e2019GB006286

"Lacking reliable estimates of past and present radiolarian diversity, we have so far been unable to determine how this vital plankton group responded to climate in the past, or to predict the impacts of anthropogenic change in the future."

I suggest toning this statement down, especially with respect to the past. Radiolarians have been used in hundreds of paleoenvironmental surveys, and although our knowledge is still very limited, we do have much information on their responses to past climate changes.

"Therefore, while radiolarian biogeography is generally partitioned into latitudinal provinces (endemic polar, cosmopolitan mid-low latitude, and tropical), there is a degree of lateral connectivity between them."

The statement about radiolarian biogeography is correct (BTW, a reference is needed here), but the authors should clarify that it specifically refers to the upper layer (~0-200 m), and the "lateral connectivity" occurs at depth.

"Range through is determined by the geologically oldest and youngest occurrence of each species. It accounts for rare taxa that may not be observed in every sample, but has the drawback of significantly underestimating richness at either end of the diversity curve"

Very much so (see: Boltovskoy D (1988) The range through method and first-last appearance data in paleontological surveys. *Journal of Paleontology*, 62:157-159). Although the range-through method modifies patterns based on point estimates strongly, in this context its use is justified. Nevertheless, the shape of the spp. richness vs. time curve is very sensitive to the proportions of non-zero records in the samples x spp. matrix. Higher proportions of zeroes produce more bell-shaped curves, whereas low ones result in shorter declines at the ends and longer plateaus in the middle. Radiolarian data, especially when the rare species are included, tend to yield matrices with >95% of blank cells, which in range-through transformations derive in rather smooth declines at the extremes, which complicates the choice of how many terminal samples should be excluded to avoid this bias. While for the SO dataset, with many samples, the range-through curve appears generally similar to the "within sample average", for the equatorial one the declines at both extremes are probably an artifact. This issue is by no means critical, but something the authors may provide more comments on, especially in view of their statement that "We thus include only robustly estimated range-through diversity spanning ~7-2 Ma"

Diversity may be interpreted as a different concept from species richness ("diversity" is normally used when referring to one of the many proposed metrics, such as Shannon-Wiener, Margalef, Simpson, etc., some of which incorporate a measure of evenness). I suggest using "species richness" throughout the text (as in the figures), as this is the metric employed in the survey (e.g., "We constructed a biodiversity curve of tropical....", line 70)

"Our findings thus suggest that SO species are unable to consistently track their preferred temperature conditions, possibly due to biotic factors or surface water hydrographic barriers, such as the polar front."

and

"This is presumably because polar species that depend on surface water conditions (e.g. those that rely on photosynthetic symbionts) cannot survive deep submergence in the tropics" Symbiont-bearing radiolarians are particularly common in the tropics, where peak radiolarian densities are often in the uppermost 100-200 m layer. In polar areas, radiolarians with autotrophic symbionts are less abundant, probably because their peak densities are usually found deeper (200-400 m) where light is limiting.

The inability to migrate to -and survive in- deeper, lower latitude waters, may be associated with their feeding requirements, which are much more limiting at depth than at the surface. See:

Wiebe PH, Boyd SH (1978) Limits of *Nematoscelis megalops* in the Northwestern Atlantic in relation to Gulf Stream cold-core rings. Part I. Horizontal and vertical distributions. *Journal of Marine Research*, 36:119-142

for an interesting example of how cold-water euphausiids expatriated to higher temperature areas migrate to lower depths following their preferred temperature regimes, but eventually starve and die from lack of food.

Captions should be provided for explaining the contents of each Supplementary table.

Fig. 1a. Please explain what the hemisphere on top of the vertical cross-section of the ocean means. Grey arrows? Shades from purple to brownish to blue?

Fig. 1c. In the modern ocean, *Siphocampe lineata* can hardly be considered a tropical species. It is clearly a northern hemisphere polar-subpolar radiolarian.

Fig. 2. The figure would be more intuitive and easier to grasp if it were rotated 90° (Recent at the top), or at least the time axis were reversed (with Recent on the right hand side). Labels are missing for the scales, and some scales fall short of covering the range of the datapoints (e.g., in 2a, species richness within sample has several values below the bottom of the scale - 350).

In conclusion, this is an interesting and solid survey, definitely novel and of interest to a wide audience, which skillfully combines paleontological and biological information to arrive at original conclusions. I therefore recommend its publication.

Number of samples where species present (total samples=100)

Proportion of cells of the species x samples table with non-zero values: 35%

Proportion of cells of the species x samples table with non-zero values: 14%

Proportion of cells of the species x samples table with non-zero values: 6%

Actual data

Range through

Actual data

Range through

Actual data

Range through

Sspecies (N=100)

Reviewer #2:
Remarks to the Author:

Review of Trubovitz, Lazarus, Renaudie & Noble manuscript
"Marine plankton show threshold extinction response to Neogene climate change"

Recommendation: Accept with minor revision.

This manuscript analyses biodiversity patterns in radiolarians throughout last 10 Ma, both at low and high latitudes, to assess whether some of the documented events were indeed extinctions or range contractions (i.e., species expanding their range to high-, or migrating to low latitudes). The main claim is that, in response to the long-term cooling trend over the Neogene, tropical radiolarians did not experience significant extinction events, while high-latitude faunas did go through a major event, with only a fraction of their taxa being able to migrate to low latitudes in order to adapt to the changed/colder environmental conditions. The authors comment how this event might be connected to crossing a certain temperature threshold, and how future climate change might impact radiolarians even more dramatically, as the migration routes to colder conditions would be precluded to them. The approach is novel in that a summary biodiversity curve for radiolarians at low latitudes did not previously exist. The authors provide it for the first time, based on a more comprehensive completeness-based approach in enumerating species, which is much more sophisticated than what routinely done (especially in studies having a focus on biostratigraphy).

The conclusions have broad-ranging implications for climate research, evolutionary studies and ecological modelling and have thus large appeal to the wider scientific community.

In my opinion, this study has great potential to influence thinking in the field, especially in terms of creating standard procedures to approach both the collection of census data, and their synthesis in a biodiversity context.

Statistical analysis is kept to a minimum, but appropriate, valid, well-documented, and thus easily reproducible.

Feedback on information provided in the reporting summary: Quite useful, especially to understand which data were newly generated for the present study, but also for a bit more detail on the techniques.

My only concern is about how convincing such a study is, and I do indeed feel that further evidence would be required to strengthen its conclusions. Let me outline what I mean by that, as it could well be that some of the points I make are due to my wrong reading and/or are relatively easy to fix by the authors:

- This work mainly relies on a comprehensive review of Southern Ocean radiolarian biodiversity (Renaudie and Lazarus, 2013), and analysis of biodiversity data derived from the NSB database both for high and low latitudes. A look at the within-sample diversity data shown in Figure 2b suggests very good data density associated with the former studies. This study adds very few new datapoints from the equatorial Pacific. Admittedly, these data have been assembled with much more comprehensive enumerating techniques compared to NSB, or other routine biostratigraphic work. New datapoints for range-through diversity boil down to only 7 (starting with a total of 13, but with 6 eliminated at both ends due to edge effects associated with the range-through approach, which would result in underestimates of biodiversity in those samples);
- I understand the site choice, outside of the EEP equatorial divergence to minimize the impact of upwelling, but there is huge potential for this study to be expanded into the EEP (exactly with the opposite rationale in mind: compare and contrast), and potentially WEP as well (if one wanted to have a go at El Niño influence on diversity on these timescales...). While these approaches might warrant coverage in separate papers, due to the complexity of their implications, I feel that the addition of at least another location within the CEP itself would be quite valuable, as the expanded coverage would

allow to test whether the observed pattern is consistent, and not a "one-off" at the chosen location;

- Sample coverage at the site. The authors mention that the sampling was performed before a final age model had been available. However, data density seems quite sparse as it is, especially with the range-through approach (the 7 datapoints mentioned above), leading to only the 7-2 Ma interval being documented. This is coupled with a relatively wide range of options in terms of what is happening when, including high extinction rates and low-latitude range contractions (~4-5 Ma), Antarctissa rise to dominance in the Southern Ocean and drastic reduction in Pielou Equitability index (7-8.5 Ma), a three-step pattern over the Neogene with a generally defined threshold of ~6°C and an extinction event spanning about 5 Ma. It would be good if these events were better constrained (better sampling), and more clearly identified/summarized (text edit);
- I also found the quite light treatment of the Neogene climate signal a bit odd, essentially reduced to a long-cooling trend with some variability... There is a lot happening over that time interval, and much of it has strong impacts on large-scale oceanic reorganizations that could potentially affect radiolarians;
- The title mentions "threshold extinction response", but the text includes very little info on how that threshold is derived. Moreover, it is simply assumed to be a valid enough model to explain the observed pattern;
- The title mentions "Marine plankton response", but it deals with one site, seven samples, one plankton group, so I find it a bit too generalizing. Some of this generalization approach can be seen in other parts of the text as well, for example when describing biodiversity in the tropics (lines 94-98): sure, the authors used a good approach, with large sample size, completeness-based enumeration protocols, smaller sieve size... but this has been done only at one location, so it seems a bit strange to extend those results to the whole tropics;
- Partially related to this, it somewhat concerns me that about 352 taxa (out of 929 taxa in total, so a substantial portion of the dataset) are not previously described forms but are instead given in open nomenclature. I understand that it's not a problem to track them consistently, but it actually is in terms of knowing their biogeographic affinity, at least in a broad sense (high/mid/low latitude), which is therefore not known... and might thus affect the interpretation of the results in terms of how many taxa did (or did not) get extinct during the various time bins, how many migrated, etc.

My suggestions would be to at least double the sample density at the location, particularly around the parts of the record where most marked change has been observed, to constrain it even better. If at all possible, for the sake of testing the significance of the patterns, increase coverage to other areas. I honestly do not mean any of these comments as detrimental to the validity of this study, more like suggestions on how to improve the approach taken, and I completely trust the authors on how much (or how little) they would like to take onboard.

Other comments

Line 54: "... Many radiolarian species do not require photosynthesis, so their distribution tends to..."
Make it more explicit, by saying: "... Many radiolarian species (with the exception of some colonial forms, carrying photosynthesizing symbionts) do not require photosynthesis, so their distribution tends to..."

Line 65: "cosmopolitan mid-low latitude"

I realize the terminology might come from Longhurst bioprovince labels, but it sounds a bit strange for a taxon to be named cosmopolitan, and at the same time be restricted to mid-low latitudes.

Line 123: "... have remained stable in the tropics despite modest changes in the physical environment"
The work cited at the end of this sentence (Tian et al., 2018, a d13C benthic record from the eastern equatorial Pacific) documents many large-scale environmental influences on the d13C signature in the EEP, including sustained reduction trends in carbon isotope signature, changes in weathering rates, a series of aborted glaciations during the Miocene-Pliocene, at least three major glaciation events

affecting Antarctica and the NH ice-sheet, and the emergence of the Panama Isthmus. While several of these have a global character and might have manifested themselves with a (relatively) muted signature in the tropics, some would have had a very strong impact on the EEP region (e.g., NHG and closing of the Panama Seaway and all the associated oceanographic changes). All this to say that I find the mention of "modest changes" a bit of a misnomer. Maybe it's only a matter of wording... as those changes could indeed be considered modest, but only by comparing their magnitude and impact to those observed at polar latitudes.

Line 125: "By contrast, the SO exhibited significantly lower radiolarian biodiversity..."

It would probably be necessary to mention that such lower diversity in polar vs low-latitude environments is a general pattern observed in the modern world for most taxa.

Lines 152-154: "This is presumably because polar species that depend on surface water conditions (e.g. those that rely on photosynthetic symbionts) cannot survive deep submergence in the tropics." I understand the argument, but one could imagine that such reliance on symbiotic photosynthesis would not be a very prevalent, or even useful, trait in radiolarians in such (very strongly seasonally light-limited) environments, compared to availability of other food/energy sources known to be linked to radiolarians, like diatoms, copepod larvae, bacteria and organic matter associated with massive, widespread algal blooms, for example. Meaning that the number of species that would be affected by such a problem would probably be relatively small.

It would also be great if the authors would consider addressing (or at least commenting on) one peculiarity of the Neogene radiolarian record, i.e. the occurrence of quite a few colonial species during some particularly warm intervals, such as the middle Miocene. These are very well-known taxa, to the point that many of them are key biostratigraphic marker species, and very unusual in their occurrence at such high latitudes, while in the modern ocean colonial radiolarians are very abundant and diversified at low-latitudes, especially in oligotrophic regions/gyres, where their association with symbionts would probably represent a distinct adaptive advantage.

Lines 162-153: "We attribute the Late Neogene SO extinction event to relatively high-magnitude polar climate change, compared to the tropics".

When exactly did this occur, this is not explicitly mentioned in the text. It also becomes confusing to follow the short argument on the causes of it, as last 10 Ma (besides the long-term cooling trend) were very eventful from a climatic point of view, as the authors also note.

Additionally, the word "relatively" might be superfluous here, as polar amplification is probably the least controversial aspect of climate change research, well known and prevalent throughout most (if not all) the Cenozoic.

Lines 168-172: "The gradual change of $\sim 3^{\circ}\text{C}$ in the EEP, as indicated by paleotemperature proxies, appears to be below the extinction sensitivity threshold of tropical radiolarians, whereas a relatively large drop of 10°C with high intermittent variability over the last 10 Ma in the SO elicited a major extinction response once temperature thresholds were surpassed."

This "somewhere between 3 and 10°C " is as close a quantification of the threshold value as attempted, with the authors describing three steps to this extinction, and coming up with a $\sim 6^{\circ}\text{C}$ threshold value. They also specify later (line 198) how the event occurred over 5 Ma.

Lines 246-247: "The horizontal purple bars indicate latitudes known for good radiolarian (silica) preservation in deep sea sediments."

Cannot see them... also unclear how these regions would be defined.

Lines 263-264: "and extinction rate is plotted in red."

Only part of this curve is traced in red.

Lines 266-268: "Extinction rate and end-member portions of ranged-through species richness are not shown in (a) due to low density of samples in the tropics and resulting computational artifacts."
Wouldn't data sparseness also similarly affect the diversity curves for the interval 5-6.5 Ma, for which very few datapoints exist?

Line 281: Figure 3.

I find this figure a bit difficult to unpack, especially as it is not very clear what the plotted "Percentage of SO species richness" actually represents. Would the very low value associated with the 7-8Ma bin indicate this time interval is characterized by very few high-latitude species? Would it suggest this is where the extinction event happened? Or maybe its precursor event, impoverishing the high latitude faunas and making them more susceptible to later extinction over next few Ma?

Lines 488-489: "Approximately half of the 929 species-level taxa observed in this study are new species awaiting formal description".

Wouldn't this be problematic in terms of describing and interpreting biodiversity and extinction trends? I understand the approach taken (document taxa "disappearing" or "migrating", based on their occurrence in age bins in high/low latitude areas), but this is a very significant proportion of the assemblage and thus probably blurs those estimates, also because the geographical adaptation of these open-nomenclature taxa is probably ill-defined/understood. In some groups, this uncertainty is quite important: for example, about 94 undetermined plagonid taxa, probably a reflection of the small mesh size used in this study (which is good practice, but it lends itself to this problem, including specimens that are usually under-reported), 61 undetermined trissocyclid taxa, 38 undetermined actinommid taxa, and 39 undetermined hexastylid taxa.

Lines 623-625: "All curves began to flatten after approximately 4000 specimens had been counted, indicating that few new species would be encountered with continued sampling."

It is very difficult to pick up a flexure/flattening point. Furthermore, curves do not really seem to have flattened yet, so the second statement is slightly misleading... for all samples, between 4000 (taxa envelope between ~350 and 410) and 5000 specimens (~360-440), about 10-30 extra taxa would still be detected with higher sample size.

Lines 529-530: typo, should read "between its first and last occurrence"

Line 534: typo, should read "it has significant"

Line 541: typo, should read "including those subject to analytical"

Comments on taxa included in Supplementary material

Preamble: the authors have world-class taxonomic expertise, so my comments, in particular on last four items in this list, might be more due to my own ignorance of recent taxonomic revisions of some genera (e.g., *Tetrapyle*/*Stylochlamidium*).

Acrosphaera arktios: Typo.

Artobotrys borealis: the figured specimen is not very typical.

Amphimelissa setosa: too many cephalic lobes, enveloped by an external layer, which is more typical for *Botrycyrtis scutum*. This ID is important as *A. setosa* is a polar species, displaying a regional extinction in the North Pacific during the Late Pleistocene.

Collosphaera tuberosa: not typical form, as pores too large and relatively regular.

Plectacantha oikiskos: Typo.

Cladoscenium ancoratum: Typo.

Pseudodictyophimus? Sp. 4: This is probably *P. gracilipes* (forma/subspecies) *bicornis*.

Actinomma medianum: Somewhat atypical, closer to *Actinomma antarcticum*.

Haliometta miocenica: correct spelling? This taxon is reported as *Haliometta* in Caulet's microtaxa genera database.

Helisoma dispar cf sp 1: Typo.

unknown actinommid >6 spines sp 4: Pretty close to *Actinomma leptodermum*. Do not agree with the differential diagnosis either, as *Hexacantium* species of this size (e.g., *H. pachydermum*) have much smaller, very regular pores and their spines (generally six, very occasionally 7 or 8) are arranged in three axes, perpendicular to each other.

Hexacantium hostile: This should be named *H. armatum-hostile* (as in WORMS, temporary name), as the *H. hostile* described by Cleve has a very different shape (rhombic outline of the cortical shell) and many short byspines (similar, but not identical, to the specimen figured as "unknown hexastylid sp 16").

Stylatractus santaeannae: Typo.

Hexacantha dodecantha: As mentioned by the authors, this is known as *Hexapyle*.

Pylodiscus spinulosus: Very similar to *Hexapyle dodecantha*...probably name changed according to Zhang and Suzuki (2017), same for the two taxa that follow, as *Tetrapyle circularis* would be *Tetrapyle octacantha*, and *Tetrapyle fruticosa* would be *Octopyle stenozone*.

unknown spongodiscid sp 1: *Stylochlamidium venustum* or *S. asteriscus*?

unknown spongodiscid sp 2: Isn't this *Spongopyle osculosa*?

Stylochlamidium venustum: broader concept, as the polygonal shape of the rings suggests *S. asteriscus*.

Reviewer #3:

Remarks to the Author:

The authors comprehensively describe findings of radiololarian assemblages from the past 10 million years at a new site from the Eastern Equatorial Pacific and compare these data with data on radiololarian assemblages catalogued from the Southern Ocean. Using a storyline of isothermal submergence pathways, the authors attempt to place their findings in a global context. They also infer an extinction 'event' although exactly when or why it occurs seems vague. Based on these ideas that are "contrary to the prediction of modeling studies", the authors then describe a scenario for the future based on what might have happened in the past.

While I appreciate the importance and novelty of these data, the authors fail in my view to properly respect the limitations in these data and compound this by regularly overdoing their conclusions. In summary, I am unconvinced of the basic extinction-event storyline, let alone the proposed implications. For example in the abstract, "the first ever full census of late Neogene tropical and cosmopolitan warm water radiolarian taxa". This seems to be a great exaggeration. The authors seem to actually describe findings from a single tropical core site, which is at most representative of the Eastern Equatorial Pacific, but certainly not all the tropics, while the subtropics and temperate zones are ignored (reliance of storyline on the isothermal submergence pathways, see below). Without respect for this limitation, this single site is then used to assume... "that ~70% of the "missing" polar radiolarians did not move equatorward but instead became extinct". The authors then describe "a threshold response to regional temperature change" scheme that to me seems arbitrary, temporally vague by the descriptions herein and by the observed extinction 'rate' (see below), and unsupported by references to established mechanisms. The above issues make me very sceptical about their attempts to predict the future. The results are otherwise interesting and with a number of adjustments, including a greater use of statistics, and with much toning down on the language and interpretations, the paper could be very interesting.

My three main issues are the following (although others in the line-by-line comments are also substantial):

--Major assumption that the "missing" species that are also not found in the tropical East Pacific must

go extinct. Why can't they go to other tropical areas? Or be unobserved in other geographical locations? To me it seems that the entire rest of the world remains for organisms to migrate to, thus avoiding or delaying extinction. Either way, the implications of this need discussing and weighing on the wording used in this MS.

--Given the consistent numbers of potential 'extinctions' (Fig. 3), no event seems evident (the authors could convince me otherwise by use of statistics and a less biased estimation of extinction rate). This is made worse by the potential Signor-Lipps effect, that the last occurrences are unlikely to be the true last individuals, which is especially relevant because of the geographical sampling gaps. Additionally, is this potential extinction rate above the temporal average of the last 10s of millions of years? A comparison of extinction rate with that of a longer time series would be useful to see if it is or is not.

--The issue of geographical sampling gaps seems to be swept aside because of the storyline's reliance on "isothermal submergence pathways", but this argument seems to have a problem too, which the authors acknowledge in L152-4. Temperature is not everything, and even though a cool-adapted species could find optimal temperatures at lower latitudes, deep water has other very different conditions to surface waters, such as heterotrophic food availability, even given the polar winter. I would expect that not all species would be able to maintain populations under these new conditions. So the high latitude species that you found in the tropical, are these just deep water cosmopolitan species or they do include some of the shallow water, previous polar endemics? Rather than just assuming this mechanism to operate perfectly for all taxa, this mechanism needs proper evaluation.

Line-by-line comments:

L46. low-mid latitude photic zone 'alone'? This is quite a large region and for most taxonomic groups this is where greatest richness is observed. What is the radiolarian latitudinal diversity gradient? Also, vague... what is 'low mid'? This goes for lots of places in this paragraph. We need some numbers in here especially for latitude, which is key to the paper

What are the taxa here? Species, genera? Sometimes the papers says species sometimes taxa. Is there a difference? I get the impression that it is only species, so stating this clearly when referring to your own results would be best.

L55 split watermass: water mass

L68: need to know what does low vs high mean here

L74: More information needed on exactly what "missing" means: unobserved?

L76: Are there strong isothermal submergence pathways from the Southern Ocean? I understood the SO to be relatively self-contained as a water body but this maybe just shallow water.

Plus the route from the SO to the Eastern Pacific is surely only one potential route, while to the S Atlantic, Indian Ocean and W Pacific would be other routes for survival?

L77: do the temperatures experienced at either latitude overlap enough to permit range shifts of species from one to the other?

L81: from a synthesis

L83. How can this statement be possibly made? Needs some qualification

L85: Avoid saying true diversity as it is unlikely we can ever judge 'true' diversity.

L84: All groups have lots of rare species and few common

L87: "fixed counts of a few hundred specimens per sample". Sounds like rarefaction, the purpose of which seems lost here (and needs adding for a balanced discussion), which is to correct richness estimates for sampling effort

89: this reference 32 is about functional structure. That is a big leap to ecosystem functioning and recovery, although other references exist for such a statement. Actually, this sentence seems unnecessary here. The next sentence's reason is sufficient and less distracting

L97: "adequate documentation of full biodiversity" what does this mean??

L99: "diversity (species richness)". Just say species richness in all cases to avoid confusion with Shannon entropy etc

L103: "averaged 355 species (+/- 19)". Mean or median? Use proper plus or minus symbol \pm . Is 19 SD, se, range? Also L110, 113, etc

L105: Or perhaps taxa that were extirpated but recolonised later?

L111: Spell out acronyms on first mentioning: NSB.

"tropical Pacific region". Some information on longitude or coastline limits would be useful too

L123: What is meant by "modest changes in the physical environment⁴¹". Your figure shows gradual cooling. Why not say "cooling" instead of "changes". Unless you mean other changes? If so, be more specific

L128: "<20% of SO radiolarian assemblages". By abundance?

L129: "nearly tripled in dominance to >40%". This doesn't make sense. Do you mean abundance or proportion? Dominance isn't the right word here. What number triples?

"without substantial change in species diversity". Without significant change? Some quantification of this statement is needed.

L130: Equitability should not be capitalised. This sentence then needs some interpretation or better linkage with the previous sentence i.e. evenness drops as Antarctissa becomes more dominant

L133-5: "Such disruptions in community structure". What exactly do you mean here? The proliferation of a single taxa? Like of "disaster taxa"? It also could just be a regional environmental change rather than assuming a mass extinction. This mentioning seems a bit too sudden without greater qualification

L137: "a cross-comparison with low-latitude species occurrences reveals that 71% of the polar diversity decline was due to extinction". I find this argument unconvincing, especially in its current strong wording. Surely there were other regions the taxa could have been extirpated to, even if some of those did eventually go extinct, or sampling reasons. These possibilities need greater attention before I would be convinced, and even then the strength of wording needs toning down.

Also the percentages need the number of species alongside them for reader confidence i.e. "71% (of 100 species)"

L142: "While 71% of the species suffered extinction". Just because your data cannot account for these species doesn't necessarily mean they went extinct. You are already aware of the Signor-Lipps effect and the big geographical holes in the sampling make this effect even more likely

L145: "cosmopolitan taxa that restricted their ranges to the tropics to avoid global extinction".

Assuming that they became restricted to the tropics just because they disappear from the polar regions doesn't seem logical. There are the subtropical and temperate regions that are completely ignored here. Why would polar extirpation threaten a cosmopolitan species with extinction? And suggesting any species response as by intention to avoid extinction sounds wrong. They avoid extinction by a response, not responding to avoid extinction. Change wording.

L149: "Significantly less". Is this statistical significance? If not, change to 'substantially' or something

L154: "found that only a small fraction of species truly migrated from one place to another (<1%".

Change 'another' to 'one other', since you do not show if those species 'truly migrated' to many places in the Southern Hemisphere.

"contrary to what modern ecological models would predict (e.g. 3,5). This response pattern shows that ecologic range migration models may not realistically scale up to evolutionary processes over long time periods. Our findings thus suggest that SO species are unable to consistently track their preferred temperature conditions, possibly due to biotic factors or surface water hydrographic barriers, such as the polar front. " This would be a very strong and profound conclusion. Since I find the evidence far from compelling, I would strongly caution against such a strong statement based on such evidence with substantial geographical holes.

L165: Meaning of 'compounded' is unclear, and in other cases

L171: "drop of 10°C with high intermittent variability over the last 10 Ma in the SO". Just stated above that this is an inferred, rather than directly estimated, drop for the SO. Please adjust to avoid any

mislead

L174: "change" or "drop"?

L175: "amount of change". Very vague. Does amount mean rate or magnitude? Does change mean oscillation (variance) or drop (mean)?

Is there any experimental evidence as to what these thresholds really are? Besides saying "apparently", which is simply acknowledging a correlation, this whole phased response scheme is arbitrary and the underlying mechanisms (besides the vague 'temperature change') are not described
L180: "loss". Does this mean local i.e. extirpation, or extinction too?

L181: "compounded temperature decrease reached 10°C below pre-threshold levels". Compounded again. Also unclear: looking at the mean line, a decline of 10 degrees from the 'pre-threshold' is only achieved when we reach modern times. But I think the extinction event you mean is earlier than that?

L187: The argument needs balance. IPCC also predicts strong impacts on tropical ecosystems.

L189: 300 years, not 200.

L193: "directional trend". This comes out of the blue. What is your hypothesis here that is different from magnitude?

L208: A reference for this evidence?

L211: Potentially at the loss of some of the tropical edge of their range?

L213: "they predict marine plankton temperature thresholds may be surpassed within the next 200 years". Your findings do not predict this. This is an oversimplification. You observe that past. These observations might inform the future, but suggesting they predict it 1:1 is unwise.

L214: "with profound consequences". Your study also does not predict these, although this sentence is written as such.

L215: "Our results document...". Your results suggest... As noted above, I find the alternative explanations for what happened to the uncounted "missing" taxa have not been dealt with properly and remain a possibility. This would weaken the 'event' hypothesis

"Major extinction event". Your results cover radiolarians. It may be an event for them (although see above) but is there any evidence this also happened for other taxonomic groups? State the event as strongly as you have convincing evidence for but not more.

L217: "only slightly lower in magnitude to biodiversity decreases seen globally in some marine plankton groups". Vague. Please be specific. Are these 'some groups' those that were less affected or more affected? Including radiolarians?

L221: Change "will" to "may". Also the following line: whether adaptation can play a role is an ongoing question and some references to that discussion are needed here.

L223: "we predict that in the next several hundred years radiolarians could suffer an extinction event". Scientific discipline needed here. What you talk of here is not a scientific prediction as your paper is not about the future. At best this is an informed opinion. The wording needs toning down or supported with studies that have indeed modeled, and thus predict, the future. I thoroughly agree that the past can be a useful guide for the future, but suggesting it predicts the future is essentially naïve.

L228: "before the extinction threshold is breached". This is a hypothetical threshold. To really declare the existence of a threshold one would need to run controlled experiments, which, of course at this scale, are impossible. Again, far more caution is needed in this message. Additionally, check the definition differences between 'breach' and 'breach'... I think you mean breached.

L231: "we urge that such records be more commonly integrated with modern climate impact studies in the future". I agree with this statement wholeheartedly, but this study, as presented, does not successfully acknowledge the limitations of such records for use in providing context for predicting the future. Science must be disciplined or we risk losing the confidence of those who use it.

FIGURES

Fig. 1: Generalized biogeographic provinces. Do these have a reference? I assume they are not just guesswork.

Fig.2A: Not clear what is going on with the y axes and which one is related to within sample richness. Consider a log axis with values unlogged to show all richnesses on a single axis.

Perhaps spell out acronym SO here so the figures are standalone

2B: Using a 1Ma moving average in plotting the line would bring out the 'broad patterns' you want the reader to restrict themselves to. What do the background shaded rectangles represent? Can you show on here unequivocally where the extinction event is?

Fig. 3: This seems a strange way of grouping the species: of the 'migrations to low latitudes', which were cosmopolitan and which were endemics? There are three things going on: extinct vs range shift vs remained, range size e.g. cosmopolitan, latitudinal preference/temperature adaptation.

Migration cannot really be quantified as it cannot include those that migrated to other regions. At most, it is just migrations to the Eastern Pacific.

These 'four possible outcomes' act like the entire world was sampled by these two regions of the EEP site and the SO

What is EEP?

Even if these do represent true extinctions, I can't see where a single event is. Is there any event that has a significantly high number of extinctions? Instead it looks like extinctions are well-spread over the past 10 million years with no significant event?

METHODS

L476: "samples yielded very well to moderately well preserved". Well- and well-preserved need hyphenating for clarity.

L510: "once visual inspection of the collection curve indicated that it was sufficiently flattened (Supplementary Figure 2)" ?? I think a little more explanation is needed here, or deleting this 'sufficiently flattened' idea, since the curves are far from flat.

L515: "are similar to those reported by Renaudie and Lazarus". What is similar? Why not just report the levels here? Perhaps some statistics would be useful

L537: Delete sentence "Therefore, we do not display the artifactual edges of the tropical diversity range-through curve in Figure 2b." The next sentence says the same thing but better.

L541: "including those with subject to". Those which were subject to?

L554: 'Extinction analysis'. There seems to be no actual analysis here

Given that the diversity estimation edge effects are quite a problem in this study, why not look for some alternative ways to show richness? Such as corrected sampled-in-bin diversity in divDyn

How is extinction rate calculated? Simply the number of species you don't see again? See the R package divDyn for alternatives.

Reviewer 1

Introductory remarks: Review of "Marine plankton show threshold extinction response to Neogene climate change" by Trubovitz, S., Lazarus, D., Renaudie, J., & Noble, P.J.

Based on analyses of radiolarian species richness changes throughout the last 10 million years at the equator and in the Southern Ocean, the authors conclude that a large proportion of the high latitude species have been unable to cope with global cooling by shifting their distributional ranges toward lower latitudes, and went extinct. Extrapolating these findings to the current global warming trend, they suggest that many living warm and temperate-water radiolarians may not be able to shift their ranges polewards or move to deeper (and cooler) layers in order to persist in their preferred thermal regime, but will also go extinct. Further, in a scenario of global warming, polar species will lack a refuge altogether, and will also conceivably disappear.

The work is definitely interesting, not only for the radiolarian community, but also for a much wider audience of biological and paleontological oceanographers, as well as for scholars interested in the effects of global warming on the biota. The materials and methods used are generally solid, and the conclusions are convincing.

I have only one major observation, and several minor suggestions.

Comment: The major point is the scarcity of datapoints for the equatorial site (13 samples, one per 0.5 Ma), which contrasts sharply with the wealth of information for the Southern Ocean. Given the exhaustive taxonomic work performed on the data, this difference probably does not invalidate the conclusions arrived at, but its potential implications deserve further justification.

Response: We have added a new sample to the eastern equatorial Pacific (EEP) time series (0.7 Ma). This had no significant effect on our results or conclusions, thus in a simple way demonstrates our results are not highly sensitive to the number of samples in our time series. We have also further explained the rich source of supportive information in the NSB database (in terms of both samples and geographic locations within the tropical Pacific), which confirms the overall trend of our findings in the EEP. Together, the additional data we collected and NSB database analysis indicate that more data from the equatorial Pacific would be unlikely to have a meaningful impact the results of our study.

Comment: BTW, I would have expected to see, in the Supplementary tables, the two complete databases with presence/absence data for each species in the EEP and in the SO.

Response: Supplementary Table 4 gives species names, mean abundance, maximum abundance, minimum abundance, and standard deviation in abundance for the EEP dataset. Supplementary Table 6 has been added to provide the same information for the SO dataset. On acceptance of this manuscript the complete datasets (including presence/absence data for each species) will be uploaded to the NSB database and thus freely accessible to other workers.

Comment: Several minor points follow.

"...contribute to the global silicon cycle..." - A more updated and more specific reference for this statement: Llopis Monferrer N, Boltovskoy D, Méndez Sandin M, Tréguer P, Not F, Leynaert A (2020) Estimating biogenic silica production of Polycystina and Phaeodaria (Rhizaria) in the global ocean. *Global Biogeochemical Cycles*, 34:e2019GB006286

Response: Suggested reference added.

Comment: "Lacking reliable estimates of past and present radiolarian diversity, we have so far been unable to determine how this vital plankton group responded to climate in the past, or to predict the impacts of anthropogenic change in the future."

I suggest toning this statement down, especially with respect to the past. Radiolarians have been used in hundreds of paleoenvironmental surveys, and although our knowledge is still very limited, we do have much information on their responses to past climate changes.

Response: Statement qualified; changed "diversity" to "species richness", and "determine" to "robustly determine." This accurately differentiates our study (based on a full species richness survey) from previous work that has primarily drawn conclusions from relative abundances of common species.

Comment: "Therefore, while radiolarian biogeography is generally partitioned into latitudinal provinces (endemic polar, cosmopolitan mid-low latitude, and tropical), there is a degree of lateral connectivity between them."

The statement about radiolarian biogeography is correct (BTW, a reference is needed here), but the authors should clarify that it specifically refers to the upper layer (~0-200 m), and the "lateral connectivity" occurs at depth.

Response: Added reference, and clarified that provinces are based on surface water.

Comment: "Range through is determined by the geologically oldest and youngest occurrence of each species. It accounts for rare taxa that may not be observed in every sample, but has the drawback of significantly underestimating richness at either end of the diversity curve"

Very much so (see: Boltovskoy D (1988) The range through method and first-last appearance data in paleontological surveys. *Journal of Paleontology*, 62:157-159). Although the range-through method modifies patterns based on point estimates strongly, in this context its use is justified. Nevertheless, the shape of the spp. richness vs. time curve is very sensitive to the proportions of non-zero records in the samples x spp. matrix. Higher proportions of zeroes produce more bell-shaped curves, whereas low ones result in shorter declines at the ends and longer plateaus in the middle. Radiolarian data, especially when the rare species are included, tend to yield matrices with >95% of blank cells, which in range-through transformations derive in rather smooth declines at the extremes, which complicates the choice of how many terminal samples should be excluded to avoid this bias. While for the SO dataset, with many samples, the range-through curve appears generally similar to the "within sample average", for the equatorial one the declines at both extremes are probably an artifact. This issue is by no means critical, but something the authors may provide more comments on, especially in view of their statement that "We thus include only robustly estimated range-through diversity spanning ~7-2 Ma"

Response: Due to the range-through edge effects, we decided to use coverage-based extrapolation to estimate total species richness, given recent advances in extrapolation methodology (e.g. Chao *et al.* 2020). We have therefore replaced range-through diversity curves with extrapolation curves in Figure 2. The suggested Boltovskoy (1988) reference was added to our discussion of range-through methods, which was used in our extinction versus extirpation analysis of SO species.

Comment: Diversity may be interpreted as a different concept from species richness ("diversity" is normally used when referring to one of the many proposed metrics, such as Shannon-Wiener,

Margalef, Simpson, etc., some of which incorporate a measure of evenness). I suggest using "species richness" throughout the text (as in the figures), as this is the metric employed in the survey (e.g., "We constructed a biodiversity curve of tropical....", line 70)

Response: Done. (A few "biodiversity" mentions were kept where intended for a more general statement that includes ecology.)

Comment: "Our findings thus suggest that SO species are unable to consistently track their preferred temperature conditions, possibly due to biotic factors or surface water hydrographic barriers, such as the polar front."

and

"This is presumably because polar species that depend on surface water conditions (e.g. those that rely on photosynthetic symbionts) cannot survive deep submergence in the tropics" Symbiont-bearing radiolarians are particularly common in the tropics, where peak radiolarian densities are often in the uppermost 100-200 m layer. In polar areas, radiolarians with autotrophic symbionts are less abundant, probably because their peak densities are usually found deeper (200-400 m) where light is limiting.

The inability to migrate to -and survive in- deeper, lower latitude waters, may be associated with their feeding requirements, which are much more limiting at depth than at the surface. See: Wiebe PH, Boyd SH (1978) Limits of *Nematoscelis megalops* in the Northwestern Atlantic in relation to Gulf Stream cold-core rings. Part I. Horizontal and vertical distributions. *Journal of Marine Research*, 36:119-142

for an interesting example of how cold-water euphausiids expatriated to higher temperature areas migrate to lower depths following their preferred temperature regimes, but eventually starve and die from lack of food.

Response: We added that both food and light limitation could be factors that prevented SO species from successfully migrating to lower latitudes. We have also added the suggested *Nematoscelis megalops* example and reference to the text.

Comment: Captions should be provided for explaining the contents of each Supplementary table.

Response: Agreed; descriptions of each Supplementary table were inputted to the *Nature Communications* online manuscript submission system with our revision, and also attached as a separate file. We defer to the editor on how best to format and include these captions with our tables.

Comment: Fig. 1a. Please explain what the hemisphere on top of the vertical cross-section of the ocean means. Grey arrows? Shades from purple to brownish to blue?

Response: Explanations added to figure caption.

Comment: Fig. 1c. In the modern ocean, *Siphocampe lineata* can hardly be considered a tropical species. It is clearly a northern hemisphere polar-subpolar radiolarian.

Response: *Siphocampe lineata* has been replaced by *Centrobotrys thermophila* as an example of a tropical species.

Comment: Fig. 2. The figure would be more intuitive and easier to grasp if it were rotated 90° (Recent at the top), or at least the time axis were reversed (with Recent on the right hand side).

Response: There is unfortunately no solution to this as subcommunities in science have picked different conventions. Our axes orientations follow the common convention used by the micropaleontology/paleoceanography community, so we have not made this change.

Comment: Labels are missing for the scales, and some scales fall short of covering the range of the datapoints (e.g., in 2a, species richness within sample has several values below the bottom of the scale - 350).

Response: Scale lengths and labels fixed.

Concluding remarks: In conclusion, this is an interesting and solid survey, definitely novel and of interest to a wide audience, which skillfully combines paleontological and biological information to arrive at original conclusions. I therefore recommend its publication.

Reviewer 2

Introductory Remarks: Note: A more "pleasing to the eye", but otherwise identical, version of these comments is in the attached Word file.

Review of Trubovitz, Lazarus, Renaudie & Noble manuscript
"Marine plankton show threshold extinction response to Neogene climate change"

Recommendation: Accept with minor revision.

This manuscript analyses biodiversity patterns in radiolarians throughout last 10 Ma, both at low and high latitudes, to assess whether some of the documented events were indeed extinctions or range contractions (i.e., species expanding their range to high-, or migrating to low latitudes). The main claim is that, in response to the long-term cooling trend over the Neogene, tropical radiolarians did not experience significant extinction events, while high-latitude faunas did go through a major event, with only a fraction of their taxa being able to migrate to low latitudes in order to adapt to the changed/colder environmental conditions. The authors comment how this event might be connected to crossing a certain temperature threshold, and how future climate change might impact radiolarians even more dramatically, as the migration routes to colder conditions would be precluded to them.

The approach is novel in that a summary biodiversity curve for radiolarians at low latitudes did not previously exist. The authors provide it for the first time, based on a more comprehensive completeness-based approach in enumerating species, which is much more sophisticated than what routinely done (especially in studies having a focus on biostratigraphy).

The conclusions have broad-ranging implications for climate research, evolutionary studies and ecological modelling and have thus large appeal to the wider scientific community. In my opinion, this study has great potential to influence thinking in the field, especially in terms of creating standard procedures to approach both the collection of census data, and their synthesis in a biodiversity context.

Statistical analysis is kept to a minimum, but appropriate, valid, well-documented, and thus easily reproducible.

Feedback on information provided in the reporting summary: Quite useful, especially to understand which data were newly generated for the present study, but also for a bit more detail on the techniques.

My only concern is about how convincing such a study is, and I do indeed feel that further evidence would be required to strengthen its conclusions. Let me outline what I mean by that, as it could well be that some of the points I make are due to my wrong reading and/or are relatively easy to fix by the authors:

Comment: - This work mainly relies on a comprehensive review of Southern Ocean radiolarian biodiversity (Renaudie and Lazarus, 2013), and analysis of biodiversity data derived from the NSB database both for high and low latitudes. A look at the within-sample diversity data shown in Figure 2b suggests very good data density associated with the former studies. This study adds very few new datapoints from the equatorial Pacific. Admittedly, these data have been assembled with much more comprehensive enumerating techniques compared to NSB, or other routine biostratigraphic work. New datapoints for range-through diversity boil down to only 7 (starting with a total of 13, but with 6 eliminated at both ends due to edge effects associated with the range-through approach, which would result in underestimates of biodiversity in those samples);

Response: In addition to adding one new sample from the EEP (0.7 Ma), we have replaced our range-through diversity curve with coverage-based extrapolation, which is not subject to edge effects. Therefore, in the revised manuscript, we have effectively doubled the number of data points in the total species richness time series (14 samples are now included, instead of the original 7).

Comment: - I understand the site choice, outside of the EEP equatorial divergence to minimize the impact of upwelling, but there is huge potential for this study to be expanded into the EEP (exactly with the opposite rationale in mind: compare and contrast), and potentially WEP as well (if one wanted to have a go at El Niño influence on diversity on these timescales...). While these approaches might warrant coverage in separate papers, due to the complexity of their implications, I feel that the addition of at least another location within the CEP itself would be quite valuable, as the expanded coverage would allow to test whether the observed pattern is consistent, and not a “one-off” at the chosen location;

Response: We agree that additional sites in the WEP and/or CEP could provide valuable insight, but this is outside the scope of our current project. The large body of literature on plankton biogeography indicates pan-tropical distribution of radiolarian species, so our site location is unlikely to be a “one-off.” We have dedicated a new subsection to our *Methods* to a discussion of this topic. Furthermore, the evaluation of our results alongside the NSB database (which includes samples from 26 sites throughout the tropical Pacific) shows that the biodiversity pattern at our site mirrors the overall trend in the tropical Pacific. For the purpose of this study, we have therefore concluded that one site is sufficient. We hope that future studies will investigate species richness in the WEP and CEP for a compare and contrast project.

Comment: - Sample coverage at the site. The authors mention that the sampling was performed before a final age model had been available. However, data density seems quite sparse as it is, especially with the range-through approach (the 7 datapoints mentioned above), leading to only the 7-2 Ma interval being documented. This is coupled with a relatively wide range of options in terms of what is happening when, including high extinction rates and low-

latitude range contractions (~4-5 Ma), *Antarctissa* rise to dominance in the Southern Ocean and drastic reduction in Pielou Equitability index (7-8.5 Ma), a three-step pattern over the Neogene with a generally defined threshold of ~6°C and an extinction event spanning about 5 Ma. It would be good if these events were better constrained (better sampling), and more clearly identified/summarized (text edit);

Response: We have added one sample to the EEP time series (0.7 Ma), and used coverage-based extrapolation instead of range-through methods to evaluate total species richness. This has enabled us to use 14 EEP data points, versus the original 7 (see revised Figure 2). With regard to the SO, this region has been very well-sampled. Our revision only includes data from SO sites with the highest-quality age models (improved in NSB since our initial submission). Therefore, the rise of *Antarctissa* and decline in equitability are very well constrained, and these trends are independent of the more coarsely sampled EEP.

Comment: - I also found the quite light treatment of the Neogene climate signal a bit odd, essentially reduced to a long-cooling trend with some variability... There is a lot happening over that time interval, and much of it has strong impacts on large-scale oceanic reorganizations that could potentially affect radiolarians;

Response: We have revised the text to acknowledge other oceanographic changes, but note that temperature change in the polar regions was the largest in magnitude.

Comment: - The title mentions “threshold extinction response”, but the text includes very little info on how that threshold is derived. Moreover, it is simply assumed to be a valid enough model to explain the observed pattern;

Response: We have rewritten our Results/Discussion to clarify the definition of the threshold and its criteria. It was determined from the observed level of temperature change when the extinction interval began, and is not intended to be a model.

Comment: - The title mentions “Marine plankton response”, but it deals with one site, seven samples, one plankton group, so I find it a bit too generalizing. Some of this generalization approach can be seen in other parts of the text as well, for example when describing biodiversity in the tropics (lines 94-98): sure, the authors used a good approach, with large sample size, completeness-based enumeration protocols, smaller sieve size... but this has been done only at one location, so it seems a bit strange to extend those results to the whole tropics;

Response: Our study includes a total of 83 samples enumerated by the coauthors (14 new samples from the EEP, and 69 samples previously enumerated from the SO), plus radiolarian data from 26 tropical sites in the NSB database. We have added extensive discussion of the known representativeness of our tropical site to the revised Methods section. In our title, we use the term “marine plankton” rather than “radiolarian” in order to avoid jargon and appeal to a broader audience. The title is also accurate in that radiolarians are a key component of marine plankton as a whole, and are closely related other important plankton groups within Rhizaria. Thus, we consider these generalizations appropriate for communicating the results and implications of our study.

Comment: - Partially related to this, it somewhat concerns me that about 352 taxa (out of 929 taxa in total, so a substantial portion of the dataset) are not previously described forms but are instead given in open nomenclature. I understand that it's not a problem to track them

consistently, but it actually is in terms of knowing their biogeographic affinity, at least in a broad sense (high/mid/low latitude), which is therefore not known... and might thus affect the interpretation of the results in terms of how many taxa did (or did not) get extinct during the various time bins, how many migrated, etc.

Response: We agree that it would be ideal if more species had been previously described and had known biogeographic affinities. However, this information is unfortunately not available for many radiolarian species, and would not impact the results of our study. We assigned species biogeographic affinities based on actual observations in our EEP samples, and the SO samples surveyed by Renaudie and Lazarus (2013). All taxonomic concepts were cross-validated by the coauthors, including forms in open nomenclature from both regions. We therefore did not rely on previous literature to determine species geographic distributions.

Comment: My suggestions would be to at least double the sample density at the location, particularly around the parts of the record where most marked change has been observed, to constrain it even better. If at all possible, for the sake of testing the significance of the patterns, increase coverage to other areas.

Response: We have added one more sample in the EEP (0.7 Ma) and found no significant change in the species richness or evenness patterns. One of the primary results of our study was remarkable evolutionary and ecological stability in the EEP over the last 10 million years, so we would not expect greater sample density to change or better constrain this result. In addition, tropical stability is confirmed by the NSB database, which includes much fewer taxa than our study but has broad coverage across the tropical Pacific. While outside the scope of this study, we agree that future work is needed to better characterize high-resolution temporal and geographic patterns among tropical radiolarians.

Comment: I honestly do not mean any of these comments as detrimental to the validity of this study, more like suggestions on how to improve the approach taken, and I completely trust the authors on how much (or how little) they would like to take onboard.

Response: We appreciate your thoughtful feedback, and have made our best effort to apply your recommendations where possible, or otherwise explain our rationale for not making a change.

Comment: Other comments

Line 54: "... Many radiolarian species do not require photosynthesis, so their distribution tends to..."

Make it more explicit, by saying: "... Many radiolarian species (with the exception of some colonial forms, carrying photosynthesizing symbionts) do not require photosynthesis, so their distribution tends to..."

Response: Changed to be more explicit, but also acknowledge that there are some solitary forms that carry photosynthesizing symbionts.

Comment: Line 65: "cosmopolitan mid-low latitude"

I realize the terminology might come from Longhurst bioprovince labels, but it sounds a bit strange for a taxon to be named cosmopolitan, and at the same time be restricted to mid-low latitudes.

Response: "Cosmopolitan" removed.

Comment: Line 123: "... have remained stable in the tropics despite modest changes in the physical environment"

The work cited at the end of this sentence (Tian et al., 2018, a $\delta^{13}\text{C}$ benthic record from the eastern equatorial Pacific) documents many large-scale environmental influences on the $\delta^{13}\text{C}$ signature in the EEP, including sustained reduction trends in carbon isotope signature, changes in weathering rates, a series of aborted glaciations during the Miocene-Pliocene, at least three major glaciation events affecting Antarctica and the NH ice-sheet, and the emergence of the Panama Isthmus. While several of these have a global character and might have manifested themselves with a (relatively) muted signature in the tropics, some would have had a very strong impact on the EEP region (e.g., NHG and closing of the Panama Seaway and all the associated oceanographic changes). All this to say that I find the mention of "modest changes" a bit of a misnomer. Maybe it's only a matter of wording... as those changes could indeed be considered modest, but only by comparing their magnitude and impact to those observed at polar latitudes.

Response: Specified that EEP changes were "relatively modest" by comparison to changes in the high latitudes. Added mention of the other environmental changes reported by Tian *et al.* 2018.

Comment: Line 125: "By contrast, the SO exhibited significantly lower radiolarian biodiversity..."

It would probably be necessary to mention that such lower diversity in polar vs low-latitude environments is a general pattern observed in the modern world for most taxa.

Response: General pattern now mentioned and referenced, as well as its relevance to modern radiolarians specifically.

Comment: Lines 152-154: "This is presumably because polar species that depend on surface water conditions (e.g. those that rely on photosynthetic symbionts) cannot survive deep submergence in the tropics."

I understand the argument, but one could imagine that such reliance on symbiotic photosynthesis would not be a very prevalent, or even useful, trait in radiolarians in such (very strongly seasonally light-limited) environments, compared to availability of other food/energy sources known to be linked to radiolarians, like diatoms, copepod larvae, bacteria and organic matter associated with massive, widespread algal blooms, for example. Meaning that the number of species that would be affected by such a problem would probably be relatively small.

Response: Done - revision emphasizes food availability as a key factor for polar species. However, the mechanism preventing survival via tropical submergence is yet unknown, so we cannot say definitively that food availability is the only or most important factor.

Comment: It would also be great if the authors would consider addressing (or at least commenting on) one peculiarity of the Neogene radiolarian record, i.e. the occurrence of quite a few colonial species during some particularly warm intervals, such as the middle Miocene. These are very well-known taxa, to the point that many of them are key biostratigraphic marker species, and very unusual in their occurrence at such high latitudes, while in the modern ocean colonial radiolarians are very abundant and diversified at low-latitudes, especially in oligotrophic regions/gyres, where their association with symbionts would probably represent a distinct adaptive advantage.

Response: This is an interesting observation, but beyond the scope of our current study and we are concerned that addressing it would distract from the main point of our paper. We will keep it in mind for possible future investigation.

Comment: Lines 162-153: “We attribute the Late Neogene SO extinction event to relatively high-magnitude polar climate change, compared to the tropics”.

When exactly did this occur, this is not explicitly mentioned in the text. It also becomes confusing to follow the short argument on the causes of it, as last 10 Ma (besides the long-term cooling trend) were very eventful from a climatic point of view, as the authors also note.

Response: In the revision we have more clearly specified that the extinction interval occurred in the SO from 5-0 Ma (Phase 3), and began after regional sea surface temperatures had dropped by $\sim 6^{\circ}\text{C}$ below pre-threshold (Phase 1) levels.

Comment: Additionally, the word “relatively” might be superfluous here, as polar amplification is probably the least controversial aspect of climate change research, well known and prevalent throughout most (if not all) the Cenozoic.

Response: We agree, but have decided to keep “relatively” to clarify our point of reference: the amount of change in the tropics.

Comment: Lines 168-172: “The gradual change of $\sim 3^{\circ}\text{C}$ in the EEP, as indicated by paleotemperature proxies, appears to be below the extinction sensitivity threshold of tropical radiolarians, whereas a relatively large drop of 10°C with high intermittent variability over the last 10 Ma in the SO elicited a major extinction response once temperature thresholds were surpassed.”

This “somewhere between 3 and 10°C ” is as close a quantification of the threshold value as attempted, with the authors describing three steps to this extinction, and coming up with a $\sim 6^{\circ}\text{C}$ threshold value. They also specify later (line 198) how the event occurred over 5 Ma.

Response: Text has now been reworded to make threshold and phase definitions more clear.

Comment: Lines 246-247: “The horizontal purple bars indicate latitudes known for good radiolarian (silica) preservation in deep sea sediments.”

Cannot see them... also unclear how these regions would be defined.

Response: Label was added to purple bars in Figure 2. These regions were defined based on deep sea sediment mapping; reference added to figure caption.

Comment: Lines 263-264: “and extinction rate is plotted in red.”

Only part of this curve is traced in red.

Response: The 0-1 Ma time bin was excluded from Figure 2 due to unavoidable edge effect on the most recent sample (likely to overestimate extinction rate). This is now explicitly stated in the figure caption, and explained in greater detail in the Methods section.

Comment: Lines 266-268: “Extinction rate and end-member portions of ranged-through species richness are not shown in (a) due to low density of samples in the tropics and resulting computational artifacts.”

Wouldn't data sparseness also similarly affect the diversity curves for the interval 5-6.5 Ma, for which very few datapoints exist?

Response: Samples in the middle of a time series are less likely to be subject to computational artifacts from the range-through operation (thus they are often referred to as “edge effects”). Regardless, in our revision we have replaced range-through diversity with a more appropriate method that does not have this computational artifact: coverage-based extrapolation. If this reviewer’s comment also refers to SO extinction rates plotted in Figure 2b, we agree that the data gap due to a local sedimentation hiatus somewhat reduces temporal resolution. However, it likely does not affect the extinction rate trend at 1 million-year resolution, as the extinction interval we identified did not begin until after 5 Ma, indicating that most SO species ranged-through the 5-6 Ma time bin and only later went extinct.

Comment: Line 281: Figure 3.

I find this figure a bit difficult to unpack, especially as it is not very clear what the plotted “Percentage of SO species richness” actually represents. Would the very low value associated with the 7-8Ma bin indicate this time interval is characterized by very few high-latitude species? Would it suggest this is where the extinction event happened? Or maybe its precursor event, impoverishing the high latitude faunas and making them more susceptible to later extinction over next few Ma?

Response: We have created a new version of Figure 3 with clear distinction between the raw number of species going extinct (bars) and overall magnitude of extinction in each interval (line). The primary y-axis label has also been made more explicit to avoid confusion: “Number of species last occurrences in SO per 1 my.”

Comment: Lines 488-489: “Approximately half of the 929 species-level taxa observed in this study are new species awaiting formal description”.

Wouldn’t this be problematic in terms of describing and interpreting biodiversity and extinction trends? I understand the approach taken (document taxa “disappearing” or “migrating”, based on their occurrence in age bins in high/low latitude areas), but this is a very significant proportion of the assemblage and thus probably blurs those estimates, also because the geographical adaptation of these open-nomenclature taxa is probably ill-defined/understood. In some groups, this uncertainty is quite important: for example, about 94 undetermined plagonid taxa, probably a reflection of the small mesh size used in this study (which is good practice, but it lends itself to this problem, including specimens that are usually under-reported), 61 undetermined trissocyclid taxa, 38 undetermined actinommid taxa, and 39 undetermined hexastylid taxa.

Response: We have cross-checked specimen images from both low and high latitude regions, including all open-nomenclature forms, to ensure that all taxonomic concepts are consistent. Therefore, within the context of this study there is no uncertainty due to undescribed taxa. The reviewer is correct in that we cannot directly link our data to published work due to differing taxonomic lists (with ours including so many undescribed species). Thus, we compare only the overall trend of our species richness curve to the NSB database, without requiring that individual species names match between datasets.

Comment: Lines 623-625: “All curves began to flatten after approximately 4000 specimens had been counted, indicating that few new species would be encountered with continued sampling.” It is very difficult to pick up a flexure/flattening point. Furthermore, curves do not really seem to have flattened yet, so the second statement is slightly misleading... for all samples, between 4000 (taxa envelope between ~350 and 410) and 5000 specimens (~360-440), about 10-30 extra taxa would still be detected with higher sample size.

Response: We have rewritten our Methods to make the procedure more clear. The revised manuscript also includes coverage values for each sample, and extrapolated richness estimates with standard error.

Comment: Lines 529-530: typo, should read “between its first and last occurrence”

Response: Done.

Comment: Line 534: typo, should read “it has significant”

Response: Done.

Comment: Line 541: typo, should read “including those subject to analytical”

Response: Done.

Comments on taxa included in Supplementary material Preamble: the authors have world-class taxonomic expertise, so my comments, in particular on last four items in this list, might be more due to my own ignorance of recent taxonomic revisions of some genera (e.g., Tetrapyle/Stylochlamidium).

Comment: Acrosphaera arktios: Typo.

Response: Corrected.

Comment: Artobotrys borealis: the figured specimen is not very typical

Response: Figured specimen replaced.

Comment: Amphimelissa setosa: too many cephalic lobes, enveloped by an external layer, which is more typical for Botryocytis scutum. This ID is important as A. setosa is a polar species, displaying a regional extinction in the North Pacific during the Late Pleistocene.

Response: Species name changed to *Amphimelissa setosa* cf, as it differs from our working definition of *Botryocytis scutum*.

Comment: Collosphaera tuberosa: not typical form, as pores too large and relatively regular

Response: This form was commonly observed in our material, so we changed its name to *Collosphaera tuberosa* cf.

Comment: Plectacantha oikiskos: Typo.

Response: Corrected.

Comment: Cladoscenum ancoratum: Typo.

Response: Corrected.

Comment: Pseudodictyophimus? Sp. 4: This is probably *P. gracilipes* (forma/subspecies) *bicornis*

Response: Name changed.

Comment: *Actinomma medianum*: Somewhat atypical, closer to *Actinomma antarcticum*.

Response: The specimens we observed appear to be more lightly silicified than specimens of *Actinomma antarcticum* illustrated by Benson (1966) and Bjørklund and Benson (2003). Therefore, in our manuscript we have changed the name of this species to *Actinomma antarcticum* cf.

Comment: *Haliometta miocenica*: correct spelling? This taxon is reported as *Haliommetta* in Caulet's microtax genera database.

Response: Corrected.

Comment: *Helisoma dispar* cf sp 1: Typo.

Response: Corrected.

Comment: unknown actinommid >6 spines sp 4: Pretty close to *Actinomma leptodermum*. Do not agree with the differential diagnosis either, as *Hexacontium* species of this size (e.g., *H. pachydermum*) have much smaller, very regular pores and their spines (generally six, very occasionally 7 or 8) are arranged in three axes, perpendicular to each other.

Response: Name changed to *Actinomma leptodermum*.

Comment: *Hexacontium hostile*: This should be named *H. armatum-hostile* (as in WORMS, temporary name), as the *H. hostile* described by Cleve has a very different shape (rhombic outline of the cortical shell) and many short byspines (similar, but not identical, to the specimen figured as "unknown hexastylid sp 16").

Response: Name changed to *Hexacontium armatum-hostile*.

Comment: *Stylatractus santaeannae*: Typo.

Response: Corrected.

Comment: *Hexacantha dodecantha*: As mentioned by the authors, this is known as *Hexapyle*.

Response: Genus name changed to *Hexapyle*.

Comment: *Pylodiscus spinulosus*: Very similar to *Hexapyle dodecantha*...probably name changed according to Zhang and Suzuki (2017), same for the two taxa that follow, as *Tetrapyle circularis* would be *Tetrapyle octacantha*, and *Tetrapyle fruticosa* would be *Octopyle stenozonea*.

Response: Yes, we are trying to use names in Zhang and Suzuki (2017) for internal consistency and because it is the newest revision of the group.

Comment: unknown spongodiscid sp 1: *Stylochlamidium venustum* or *S. asteriscus*?

Response: Differential diagnosis added. Our species differs in that it has irregular segments; it lacks full concentric rings around the center of the shell.

Comment: unknown spongodiscid sp 2: Isn't this *Spongopyle osculosa*?

Response: Name changed to *Spongopyle osculosa*.

Comment: *Stylochlamidium venustum*: broader concept, as the polygonal shape of the rings suggests *S. asteriscus*.

Response: Category name changed to *Stylochlamidium venustum-asteriscus*.

Reviewer 3

Introductory remarks: The authors comprehensively describe findings of radiololarian assemblages from the past 10 million years at a new site from the Eastern Equatorial Pacific and compare these data with data on radiololarian assemblages catalogued from the Southern Ocean. Using a storyline of isothermal submergence pathways, the authors attempt to place their findings in a global context. They also infer an extinction 'event' although exactly when or why it occurs seems vague. Based on these ideas that are "contrary to the prediction of modeling studies", the authors then describe a scenario for the future based on what might have happened in the past.

Comment: While I appreciate the importance and novelty of these data, the authors fail in my view to properly respect the limitations in these data and compound this by regularly overdoing their conclusions. In summary, I am unconvinced of the basic extinction-event storyline, let alone the proposed implications. For example in the abstract, "the first ever full census of late Neogene tropical and cosmopolitan warm water radiolarian taxa". This seems to be a great exaggeration. The authors seem to actually describe findings from a single tropical core site, which is at most representative of the Eastern Equatorial Pacific, but certainly not all the tropics, while the subtropics and temperate zones are ignored (reliance of storyline on the isothermal submergence pathways, see below). Without respect for this limitation, this single site is then used to assume... "that ~70% of the "missing" polar radiolarians did not move equatorward but instead became extinct". The authors then describe "a threshold response to regional temperature change" scheme that to me seems arbitrary, temporally vague by the descriptions herein and by the observed extinction 'rate' (see below), and unsupported by references to established mechanisms. The above issues make me very sceptical about their attempts to predict the future. The results are otherwise interesting and with a number of adjustments, including a greater use of statistics, and with much toning down on the language and interpretations, the paper could be very interesting.

Response: This reviewer raises important points, which we now fully address in the revised manuscript and in our responses to individual comments below. To address the reviewer's main concern regarding site choice, we have provided extensive discussion of the known pan-tropical distribution pattern for the vast majority of tropical radiolarian species (and other rhizarians, such as the planktonic foraminifera) in our revised Methods section. Thus, there is no expectation that species occurrence lists would substantially differ at other tropical locations, although different relative abundances would be probable. According to Boltovskoy *et al.* (2010), World Atlas of Distribution of Recent Polycystina (Radiolaria): "None of the species

covered was scarce in the Pacific but abundant in either the Atlantic or the Indian oceans.” Only 5 species were not present in all three ocean basins, and all of these were absent from the Indian ocean only (Boltovskoy *et al.* 2010: Table 2 and Table 3). These observations were based on the biogeographic distribution of 307 radiolarian taxa across the Atlantic (982 samples), Indian (698 samples) and Pacific (2953 samples) ocean basins. Based on this body of previous work, we believe it is reasonable to assume that a single site in the tropical Pacific is representative of the global tropical radiolarian biome. We have further validated this assumption by comparing the trend of our species richness curve to the NSB database, which includes 1541 samples from 26 sites in the tropical Pacific. As both curves showed stability in radiolarian diversity over the last 10 Ma, this suggests that our site is not aberrant and instead is representative of the tropical radiolarian biome.

Regarding the inclusion of subtropical and temperate zones, we have cited multiple studies that show isothermal submersion connects these biomes to the tropics at depth. This concept was explicitly stated to be “correct” by Reviewer 1, and was not mentioned by Reviewer 2. Further evidence was given in a recent study that found both the tropics and subtropics are characterized by the same set of radiolarian taxa (Boltovskoy and Correa, 2019). Furthermore, this study found that the bi-subpolar and transitional (temperate) zones are characterized by the following taxa: *Botryostrobus aquilonaris* (Baily, 1856), *Stylochlamydidium venustum* (Baily, 1856), *Anomalocantha dentata* (Mast, 1910), *Cenosphaera spp.*, and *Stichopilium bicornis* (Haeckel, 1887). All of these taxa considered characteristic of bi-subpolar and transitional zones were observed in our EEP samples (with some uncertainty regarding *Cenosphaera spp.* as it was not listed at the species level). While more work that includes exhaustive taxonomic surveying is warranted to confirm this pattern, existing work provides strong evidence that the temperate latitudes are not home to a unique set of taxa that are absent from the tropics. See: Boltovskoy, D. and Correa, N., 2019. Worldwide Distribution Patterns of the Planktonic Shelled Protists Radiolaria (Polycystina) and Foraminifera: Similarities and Contrasts. *Paleontological Journal*, 53(8), pp.768-773. Therefore, for the purpose of this first-order analysis, we consider one tropical site to be sufficient. The idea of pan-tropical-subtropical radiolarian species distributions is fundamental to understanding our study, so we are grateful that this reviewer brought to our attention that further explanation was needed.

In response to the reviewer’s concern about the temperature threshold, we have rewritten the text to more clearly define our criteria and now more accurately describe the extinction response as an “interval” rather than as an “event.” We appreciate the reviewer’s suggestion to test the statistical significance of the extinction interval, and believe that the statistical test results included in our revision have greatly strengthened the manuscript. We found that extinction rates <5Ma were significantly higher than at any other time during the Neogene (Wald Test; F-value: 24.43556, p-value: 0.0002). We also found that the diversity trend during Phase 3 was significantly different from slope=0 (linear regression p-values: 8.55e-10 and 3.30e-5 for raw and extrapolated species richness trends, respectively), whereas the linear regression was not significant during Phase 2 (p-values = 0.92 and 0.36 for raw and extrapolated species richness trends, respectively). We consider this to be convincing evidence that the extinction response is not defined arbitrarily, and has clear statistical support.

Comment:

My three main issues are the following (although others in the line-by-line comments are also substantial):

--Major assumption that the “missing” species that are also not found in the tropical East Pacific must go extinct. Why can’t they go to other tropical areas? Or be unobserved in other geographical locations? To me it seems that the entire rest of the world remains for organisms

to migrate to, thus avoiding or delaying extinction. Either way, the implications of this need discussing and weighing on the wording used in this MS.

Response: We have answered these questions in our response to the previous comment above. Our revised manuscript includes detailed discussion of previous work indicating that radiolarian species tend to have pan-tropical distribution in the surface water across all major ocean basins, and that the tropics include temperate and subpolar radiolarian assemblages at depth. Therefore, there is strong evidence that the “missing” SO species did not relocate to an unobserved geographic location to avoid extinction. We agree that future work in other regions would help confirm our results, but we do not consider this work necessary for validating the results of our current study.

Comment: --Given the consistent numbers of potential ‘extinctions’ (Fig. 3), no event seems evident (the authors could convince me otherwise by use of statistics and a less biased estimation of extinction rate). This is made worse by the potential Signor-Lipps effect, that the last occurrences are unlikely to be the true last individuals, which is especially relevant because of the geographical sampling gaps.

Additionally, is this potential extinction rate above the temporal average of the last 10s of millions of years? A comparison of extinction rate with that of a longer time series would be useful to see if it is or is not.

Response: We have revised Figure 3 so that it more clearly shows the number of extirpated species that went extinct, contracted range, or migrated, and the relative magnitude of extinction within each 1 million-year time bin. Extinction rate is shown in Figure 2b, not in Figure 3. We calculated extinction rate based on Foote’s boundary-crosser approach, which has been demonstrated to be the most reliable method for paleontological occurrence data (see Methods). Our revised manuscript also includes statistical analyses that show significant increase in extinction rate and decline in species richness post-5Ma, even when the most recent bin is excluded (see Methods). We agree that our original submission used poor word choice in describing extinction as an “event,” because elevated extinction was sustained over a long time period (~5 million years). Therefore, in the revision we refer to the extinction response as an “interval” (Phase 3) in order to avoid confusion.

In response to the question regarding average extinction rates: Yes, in our revision we have included a Wald test to show that extinction rates in the last 5 million years were significantly higher than at any other time in the last 22 million years (see Methods). The SO data used to perform this test were previously published by Renaudie and Lazarus (2013).

Comment: --The issue of geographical sampling gaps seems to be swept aside because of the storyline’s reliance on “isothermal submergence pathways”, but this argument seems to have a problem too, which the authors acknowledge in L152-4. Temperature is not everything, and even though a cool-adapted species could find optimal temperatures at lower latitudes, deep water has other very different conditions to surface waters, such as heterotrophic food availability, even given the polar winter. I would expect that not all species would be able to maintain populations under these new conditions. So the high latitude species that you found in the tropical, are these just deep water cosmopolitan species or they do include some of the shallow water, previous polar endemics? Rather than just assuming this mechanism to operate perfectly for all taxa, this mechanism needs proper evaluation.

Response: We agree that temperature is not everything, but it has been shown to be the primary factor controlling radiolarian distributions in the modern ocean (e.g. Boltovskoy and Correa, 2019). Furthermore, temperature is commonly the only parameter used in studies that

model the ecologic responses of plankton to climate change; thus, our goal to test these ecologic models using the fossil record is best achieved using temperature as well. To account for other possibilities, we have added the statement that polar species may be unable to migrate due to lack of food and/or sunlight at depth. However, we are only speculating as to the mechanism limiting the migration of polar species. The intention of our study is to document the pattern of extinction; specific reasons why species went extinct and could not migrate would be an interesting avenue of future research. Figure 3 and Supplemental Table 3 break down the number of SO extirpations by endemic versus cosmopolitan species. In total, we found that 55 cosmopolitan species contracted their ranges to include only the low latitudes, 12 endemic SO species migrated from the high to low latitudes, and 149 endemic SO species went extinct. Unfortunately, not enough is known about radiolarian ecology to determine the depth at which many of the endemic SO species lived.

Comment:

Line-by-line comments:

L46. low-mid latitude photic zone 'alone'? This is quite a large region and for most taxonomic groups this is where greatest richness is observed. What is the radiolarian latitudinal diversity gradient?

Response: Text has been modified to make limitations clearer. There is significant radiolarian diversity at depth, which was not sampled for DNA analyses by Tara Oceans. Substantial diversity is also present at high latitudes, although in general radiolarian diversity decreases with increasing latitude. We note the radiolarian latitudinal diversity gradient in the text and reference Boltovskoy and Correa (2019).

Comment: Also, vague... what is 'low mid'? This goes for lots of places in this paragraph. We need some numbers in here especially for latitude, which is key to the paper

Response: Latitude was removed from this statement; the high latitudes were included in Tara Oceans, but poorly sampled.

Comment: What are the taxa here? Species, genera? Sometimes the papers says species sometimes taxa. Is there a difference? I get the impression that it is only species, so stating this clearly when referring to your own results would be best.

Response: We have replaced "taxa" with "species" in most instances. On occasion, "taxa" is more appropriate, such as when discussing DNA results or other concepts beyond our dataset.

Comment: L55 split watermass: water mass

Response: Done.

Comment: L68: need to know what does low vs high mean here

Response: Done – defined low latitude as EEP and high latitude as SO in the following sentence.

Comment: L74: More information needed on exactly what "missing" means: unobserved?

Response: Sentence rewritten to be more clear.

Comment: L76: Are there strong isothermal submergence pathways from the Southern Ocean? I understood the SO to be relatively self-contained as a water body but this maybe just shallow water.

Response: The Southern Ocean does have strong isothermal submergence pathways, which are very well documented in the literature. We have cited papers specific to radiolarian-related isothermal submergence, but this is also a well-known phenomenon in geochemistry as it is responsible for the transport of nutrients that supply low latitude surface waters.

Comment: Plus the route from the SO to the Eastern Pacific is surely only one potential route, while to the S Atlantic, Indian Ocean and W Pacific would be other routes for survival?

Response: The reviewer is correct that other pathways exist and submersion to the EEP is not the only possible route. However, this is not relevant for our study given circumglobal biogeographic patterns, and geologic timescales that allow for ocean mixing.

Comment: L77: do the temperatures experienced at either latitude overlap enough to permit range shifts of species from one to the other?

Response: Yes. The vertical temperature profile in tropics overlaps fully with the poles, at depth of isothermal layer.

Comment: L81: from a synthesis

Response: Changed to “syntheses” – there have been multiple.

Comment: L83. How can this statement be possibly made? Needs some qualification

Response: It is not clear to us what the reviewer objects to here. If again this is an argument on geographic endemism, we have already addressed this concern above. If the reviewer is questioning whether we know what others have published on low latitude radiolarian assemblages - yes, we do know what has been published rather well. The senior author (Lazarus) has read or lived through the vast majority of all publications on this biome, and our statement is also backed by searches of literature archives. If the reviewer knows of literature contradicting this claim we will be happy to modify this statement.

Comment: L85: Avoid saying true diversity as it is unlikely we can ever judge ‘true’ diversity.

Response: Fixed.

Comment: L84: All groups have lots of rare species and few common

Response: That the phenomenon is common does not mean it is incorrect or irrelevant for collection of radiolarian data.

Comment: L87: “fixed counts of a few hundred specimens per sample”. Sounds like rarefaction, the purpose of which seems lost here (and needs adding for a balanced discussion), which is to correct richness estimates for sampling effort

Response: This is a description of methods in most previous radiolarian work, not the methods we used. We have added text to clarify that previous studies often employed this kind of data collection for the purpose of paleoenvironmental reconstruction, not for rarefied diversity estimates.

Comment: 89: this reference 32 is about functional structure. That is a big leap to ecosystem functioning and recovery, although other references exist for such a statement. Actually, this sentence seems unnecessary here. The next sentence's reason is sufficient and less distracting

Response: Sentence removed.

Comment: L97: "adequate documentation of full biodiversity" what does this mean??

Response: Text changed to clarify.

Comment: L99: "diversity (species richness)". Just say species richness in all cases to avoid confusion with Shannon entropy etc

Response: "Diversity" replaced with "species richness" everywhere appropriate.

Comment: L103: "averaged 355 species (+/- 19)". Mean or median? Use proper plus or minus symbol \pm . Is 19 SD, se, range? Also L110, 113, etc

Response: Fixed.

Comment: L105: Or perhaps taxa that were extirpated but recolonised later?

Response: Our study already accounted for this, as we looked at time series up to the recent. We used range-through data to evaluate extinction, so short gaps in taxon presence would not impact the analysis.

Comment: L111: Spell out acronyms on first mentioning: NSB.

Response: Done.

Comment: "tropical Pacific region". Some information on longitude or coastline limits would be useful too

Response: Done – clarified that we included all longitudes within $\leq 20^\circ$ N/S in the Pacific ocean.

Comment: L123: What is meant by "modest changes in the physical environment⁴¹". Your figure shows gradual cooling. Why not say "cooling" instead of "changes". Unless you mean other changes? If so, be more specific

Response: Changed to just temperature.

Comment: L128: "<20% of SO radiolarian assemblages". By abundance?

Response: Yes; added "abundance" to that sentence.

Comment: L129: “nearly tripled in dominance to >40%”. This doesn’t make sense. Do you mean abundance or proportion? Dominance isn’t the right word here. What number triples?

Response: Same sentence as above – changed “dominance” to “abundance.”

Comment: “without substantial change in species diversity”. Without significant change? Some quantification of this statement is needed.

Response: In our revision we have fitted linear regression models to Phase 2 and Phase 3 species richness trends in order to test whether their slopes are statistically different from 0. Phase 2 did not have a slope significantly different from 0, indicating no substantial (or statistically significant) change in species richness. This contrasts with Phase 3, which did have a slope statistically different from 0. The results of these tests are now mentioned in our main text.

Comment: L130: Equitability should not be capitalised. This sentence then needs some interpretation or better linkage with the previous sentence i.e. evenness drops as *Antarctissa* becomes more dominant

Response: Fixed – sentences now linked.

Comment: L133-5: “Such disruptions in community structure”. What exactly do you mean here? The proliferation of a single taxa? Like of “disaster taxa”? It also could just be a regional environmental change rather than assuming a mass extinction. This mentioning seems a bit too sudden without greater qualification

Response: Text changed to be more specific.

Comment: L137: “a cross-comparison with low-latitude species occurrences reveals that 71% of the polar diversity decline was due to extinction”. I find this argument unconvincing, especially in its current strong wording. Surely there were other regions the taxa could have been extirpated to, even if some of those did eventually go extinct, or sampling reasons. These possibilities need greater attention before I would be convinced, and even then the strength of wording needs toning down. Also the percentages need the number of species alongside them for reader confidence i.e. “71% (of 100 species)”

Response: Text changed to clarify that 71% of the 233 lost polar species went extinct. The geographic argument of the reviewer is answered in responses to previous comments above.

Comment: L142: “While 71% of the species suffered extinction”. Just because your data cannot account for these species doesn’t necessarily mean they went extinct. You are already aware of the Signor-Lipps effect and the big geographical holes in the sampling make this effect even more likely

Response: Here again the reviewer invokes a geographic argument, which we dealt with in previous comment responses. The Signor-Lipps effect seems irrelevant here as this effect is normally invoked to explain the smearing of a sharp extinction event signal backwards in time. We do not see or claim such a pattern in this paper.

Comment: L145: “cosmopolitan taxa that restricted their ranges to the tropics to avoid global extinction”. Assuming that they became restricted to the tropics just because they disappear from the polar regions doesn’t seem logical. There are the subtropical and temperate regions that are completely ignored here. Why would polar extirpation threaten a cosmopolitan species with extinction? And suggesting any species response as by intention to avoid extinction sounds wrong. They avoid extinction by a response, not responding to avoid extinction. Change wording.

Response: Sentence reworded for clarity; “avoid extinction” removed. This is the same geographic argument we have addressed in previous responses. We have provided evidence that it is logical to assume that species that survived extirpation from the SO would be present in the tropics.

Comment: L149: “Significantly less”. Is this statistical significance? If not, change to ‘substantially’ or something

Response: Changed to “substantially.”

Comment: L154: “found that only a small fraction of species truly migrated from one place to another (<1%”. Change ‘another’ to ‘one other’, since you do not show if those species ‘truly migrated’ to many places in the Southern Hemisphere.

Response: Sentence fixed to make clear that we are referring to migration from endemic distribution at high latitudes, to endemic distribution at low latitudes.

Comment: “contrary to what modern ecological models would predict (e.g. 3,5). This response pattern shows that ecologic range migration models may not realistically scale up to evolutionary processes over long time periods. Our findings thus suggest that SO species are unable to consistently track their preferred temperature conditions, possibly due to biotic factors or surface water hydrographic barriers, such as the polar front. “ This would be a very strong and profound conclusion. Since I find the evidence far from compelling, I would strongly caution against such a strong statement based on such evidence with substantial geographical holes.

Response: The reviewer is apparently challenging this statement because s/he does not accept validity of our results. We have qualified the statement to just SO radiolarian species.

Comment: L165: Meaning of ‘compounded’ is unclear, and in other cases

Response: Changed to “total.”

Comment: L171: “drop of 10°C with high intermittent variability over the last 10 Ma in the SO”. Just stated above that this is an inferred, rather than directly estimated, drop for the SO. Please adjust to avoid any mislead

Response: Added that this temperature record is “inferred” for the SO based on stacked temperature data from the middle and high latitudes globally.

Comment: L174: “change” or “drop”?

Response: Changed to “drop.”

Comment: L175: “amount of change”. Very vague. Does amount mean rate or magnitude? Does change mean oscillation (variance) or drop (mean)?

Response: Changed to “magnitude and rate.”

Comment: L175: “amount of change”. Very vague. Does amount mean rate or magnitude? Does change mean oscillation (variance) or drop (mean)?

Is there any experimental evidence as to what these thresholds really are? Besides saying “apparently”, which is simply acknowledging a correlation, this whole phased response scheme is arbitrary and the underlying mechanisms (besides the vague ‘temperature change’) are not described

Response: We do not know of any published experimental data on radiolarian temperature tolerances. The goal of this paper is not to discover underlying mechanisms, though we agree it would be very interesting and hopefully will be addressed in future research. We have now added text making clear that our 3-phase scheme is indeed a convenience to describe the pattern, but it is based on objective events in the data: changes in evenness and extinction rate at certain points in time as temperature decreased.

Comment: L180: “loss”. Does this mean local i.e. extirpation, or extinction too?

Response: “Loss” refers to both extirpation and extinction. Our objective was to quantify how much SO species loss is attributable to extirpation versus extinction.

Comment: L181: “compounded temperature decrease reached 10°C below pre-threshold levels”. Compounded again. Also unclear: looking at the mean line, a decline of 10 degrees from the ‘pre-threshold’ is only achieved when we reach modern times. But I think the extinction event you mean is earlier than that?

Response: Text changed to be clear that extinction interval began ~5Ma, when temperature reached 6°C below threshold levels. Elevated extinction continued as temperature dropped to 10°C below pre-threshold levels, up until the recent.

Comment: L187: The argument needs balance. IPCC also predicts strong impacts on tropical ecosystems.

Response: Specified that IPCC predicts that impacts will be especially strong in polar regions.

Comment: L189: 300 years, not 200.

Response: Fixed.

Comment: L193: “directional trend”. This comes out of the blue. What is your hypothesis here that is different from magnitude?

Response: The directional trend refers to anthropogenic warming versus Neogene cooling. This is the concept we highlighted in Figure 1. We have now clarified this in the revised text.

Comment: L208: A reference for this evidence?

Response: Changed “adapt” statement to “not substantially affected” – a pattern that is evidenced by our own data.

Comment: L211: Potentially at the loss of some of the tropical edge of their range?

Response: This is an interesting question, but as we do not know the upper temperature tolerance limit for species already living in the warmest part of the modern ocean temperature gradient, we cannot predict their response.

Comment: L213: “they predict marine plankton temperature thresholds may be surpassed within the next 200 years”. Your findings do not predict this. This is an oversimplification. You observe that past. These observations might inform the future, but suggesting they predict it 1:1 is unwise.

Response: Changed “predict” to “suggest.”

Comment: L214: “ with profound consequences”. Your study also does not predict these, although this sentence is written as such.

Response: Same sentence as above – changed “predict” to “suggest.”

Comment: L215: “Our results document...”. Your results suggest... As noted above, I find the alternative explanations for what happened to the uncounted “missing” taxa have not been dealt with properly and remain a possibility. This would weaken the ‘event’ hypothesis

Response: Our statement here is justifiable given that we believe to have adequately documented basis in revision for our analysis and interpretation, particularly with regard to the biogeographic concerns of the reviewer.

Comment: “Major extinction event”. Your results cover radiolarians. It may be an event for them (although see above) but is there any evidence this also happened for other taxonomic groups? State the event as strongly as you have convincing evidence for but not more.

Response: Specified that we are referring to radiolarians.

Comment: L217: “only slightly lower in magnitude to biodiversity decreases seen globally in some marine plankton groups”. Vague. Please be specific. Are these ‘some groups’ those that were less affected or more affected? Including radiolarians?

Response: We have now added which plankton groups we are referring to. The sentence already states that the magnitude of Neogene SO extinction was lower in magnitude than the referenced End Cretaceous extinctions.

Comment: L221: Change “will” to “may”. Also the following line: whether adaptation can play a role is an ongoing question and some references to that discussion are needed here.

Response: Done. Reference added that states macroevolutionary change in marine plankton occurs on the scale of millions of years.

Comment: L223: “we predict that in the next several hundred years radiolarians could suffer an extinction event”. Scientific discipline needed here. What you talk of here is not a scientific

prediction as your paper is not about the future. At best this is an informed opinion. The wording needs toning down or supported with studies that have indeed modeled, and thus predict, the future. I thoroughly agree that the past can be a useful guide for the future, but suggesting it predicts the future is essentially naïve.

Response: We changed “will” to “could” and “predict” to “suggest.” However, we note that a prediction can be based on observational scientific evidence other than a model output. Also, predictions from models are only as good as the knowledge of the systems encoded in them. We feel that current knowledge of plankton biology is based on very time-limited data that likely misses many long-term processes. Using their outputs to make predictions about no-analog future conditions can thus also be naïve. We hope that use of both paleontologic and biologic data will help compensate for each data type's limitations.

Comment: L228: “before the extinction threshold is breached”. This is a hypothetical threshold. To really declare the existence of a threshold one would need to run controlled experiments, which, of course at this scale, are impossible. Again, far more caution is needed in this message. Additionally, check the definition differences between ‘breach’ and ‘breach’... I think you mean breached.

Response: Changed to “hypothetical threshold”; typo fixed.

Comment: L231: “we urge that such records be more commonly integrated with modern climate impact studies in the future”. I agree with this statement wholeheartedly, but this study, as presented, does not successfully acknowledge the limitations of such records for use in providing context for predicting the future. Science must be disciplined or we risk losing the confidence of those who use it.

Response: We changed “provide the necessary” to a more cautious “can provide an important.” The reviewer correctly is noting that no past [or current] history is a perfect match to future, thus not automatically an adequate basis for future prediction.

Comment: FIGURES

Fig. 1: Generalized biogeographic provinces. Do these have a reference? I assume they are not just guesswork.

Response: References added.

Comment: Fig.2A: Not clear what is going on with the y axes and which one is related to within sample richness. Consider a log axis with values unlogged to show all richnesses on a single axis.

Response: Added labels to all axes, and shifted positions of datasets so that richness curves are more readable.

Comment: Perhaps spell out acronym SO here so the figures are standalone

Response: Done.

Comment: 2B: Using a 1Ma moving average in plotting the line would bring out the ‘broad patterns’ you want the reader to restrict themselves to. What do the background shaded rectangles represent? Can you show on here unequivocally where the extinction event is?

Response: We added a lowess smoother to the extrapolated SO richness data to emphasize broad pattern. Text has been added to the figure caption to explain the blue shading of the background in Figure 2b. The extinction interval is shown and labelled as “Phase 3.”

Comment: Fig. 3: This seems a strange way of grouping the species: of the ‘migrations to low latitudes’, which were cosmopolitan and which were endemics? There are three things going on: extinct vs range shift vs remained, range size e.g. cosmopolitan, latitudinal preference/temperature adaptation.

Response: Caption text has been revised to clarify categories. Migrations = movement of endemic SO species; Range restrictions = contraction of cosmopolitan species ranges to only the low latitudes.

Comment: Migration cannot really be quantified as it cannot include those that migrated to other regions. At most, it is just migrations to the Eastern Pacific.

Response: This is the same geographic objection as in other parts of the manuscript – we have already responded to this issue above.

Comment: These ‘four possible outcomes’ act like the entire world was sampled by these two regions of the EEP site and the SO

Response: Changed to “four measurable outcomes.” This is the same geographic objection as in other parts of the manuscript – we have already responded to this issue above.

Comment: What is EEP?

Response: Unclear what the reviewer is objecting to here. The text properly defines EEP at first use already.

Comment: Even if these do represent true extinctions, I can’t see where a single event is. Is there any event that has a significantly high number of extinctions? Instead it looks like extinctions are well-spread over the past 10 million years with no significant event?

Response: We have defined the extinction interval as 5Ma-recent, a period when extinction rates were significantly higher than at any other time during the Neogene (see Methods). Figure 3 is intended to illustrate that both the number of extinctions and the relative magnitude of extinction was elevated during the last 5 million years.

Comment: METHODS

L476: “samples yielded very well to moderately well preserved”. Well- and well-preserved need hyphenating for clarity.

Response: Done.

Comment: L510: “once visual inspection of the collection curve indicated that it was sufficiently flattened (Supplementary Figure 2)” ?? I think a little more explanation is needed here, or deleting this ‘sufficiently flattened’ idea, since the curves are far from flat.

Response: Better explanation added to the Methods section.

Comment: L515: “are similar to those reported by Renaudie and Lazarus”. What is similar? Why not just report the levels here? Perhaps some statistics would be useful

Response: Coverage values are now provided for both EEP and SO datasets as Supplementary tables.

Comment: L537: Delete sentence “Therefore, we do not display the artifactual edges of the tropical diversity range-through curve in Figure 2b.” The next sentence says the same thing but better.

Response: Issue no longer present, as range-through curves were removed from Figure 2.

Comment: L541: “including those with subject to”. Those which were subject to?

Response: Typo corrected.

Comment: L554: ‘Extinction analysis’. There seems to be no actual analysis here

Response: “Analysis” changed to “calculation.”

Comment: Given that the diversity estimation edge effects are quite a problem in this study, why not look for some alternative ways to show richness? Such as corrected sampled-in-bin diversity in divDyn

Response: Suggestion accepted; the revised manuscript now uses extrapolated richness rather than range-through.

Comment: How is extinction rate calculated? Simply the number of species you don’t see again? See the R package divDyn for alternatives.

Response: We have explained that extinction rate was calculated using the concept of Foote’s boundary-crossers, which we have determined to be the most appropriate method for our type of data (see Methods).

Reviewers' Comments:

Reviewer #1:

Remarks to the Author:

The authors satisfactorily addressed all the issues raised in my review. My recommendation is ACCEPT FOR PUBLICATION.

Reviewer #2:

Remarks to the Author:

After reading the comments by the other two reviewers, the arguments put forward by the authors in their rebuttal letter, and the changes they adopted to address the concerns raised, I am happy with the revised manuscript.

The main issues were around the alternative potential extirpation locations to mid latitudes, the ability of the new EEP record to capture low-latitude conditions, the sparseness of data mainly linked to the range-through technique (which the authors replaced with a more viable one, as also suggested by reviewer #3, thus resulting in doubling their sample number), the unclarity of the time and magnitude of the extinction event (now better documented and illustrated, and referred to as an interval), a more rigorous treatment of the discussed trends, and further explanation on the climatic threshold triggering the response.

All these, along with a few other minor concerns, have been successfully addressed by the authors.

Reviewer #3:

Remarks to the Author:

I congratulate the authors on a substantially improved paper and a much more convincing storyline. Being perhaps representative of a non-planktonic ecologist reader, I am now convinced about the validity of the observed extinctions, and extinction interval, taking place in the past 5 million years based on the data, analysis and logic presented. I do have some relatively minor remaining issues and addressing these should in no way diminish the importance of the work, but should add to its clarity and thus its utility:

L16. 'Permanently' is unnecessary. Extinction is always permanent

L18. Should 'predicted' be 'assumed' here?

The species undergo restructuring? ...or the assemblage undergoes restructuring?

"data from a single section can in principle give a reasonable approximation of radiolarian species occurrences and richness over time throughout a given biome." In theory, this could be quantitatively supported by data: Take a random well-sampled single core from a well-sampled latitudinal province and see how well (on average) it demonstrates the province species inventory?

L80. Not 'determine'. Some more cautious word would be wise. E.g. "Estimate"

L93. "has recorded (almost) all species preserved" Not clear how this relates to previous sentence about fossil tropical species richness. All species preserved means what? In the world? In a sample? Do most studies not identify all species in a single sample?

L132. Add 'tropical' paleocommunities

L147 and 154: Confusing way to report these statistics e.g, "linear regression two-sided t-test p-values = 0.92 and 0.36 on 9 degrees of freedom for raw and extrapolated species richness trends from 8.5-5Ma, respectively). ". Perhaps say two-sided t-test of linear regression coefficients? Nine degrees of freedom is very few. Not clear why there is so great a difference vs line 154? Would be nice

to see the coefficients/means being compared too, also for line 154

L152: "Many of the species marginalized by the rise in Antarctissa " Can we really say this? Perhaps Antarctissa just made use of niche space already vacated by the other species, or their niches did not overlap at all and the rise and falls are simple correlations responding to some other driver?

L162: 71% of the species losses were...: 'Was' to 'were', or just 'went extinct'. So that the reader understands your logic, perhaps add the justification here to, "71% of of the 233 polar species lost were not observed in coeval or younger low-latitude sediments, so we infer that they went extinct".

L183: "233 SO species went extinct" I thought 71% of this number went extinct? Or is this number referring to a different thing than the previous '233' (L162)?

L193: "the SO species impacted by high-latitude cooling": Is this of the 233 species lost again? Or simply all of the species that were polar at the time?

L199: "as they submerged ultimately died". 'And' ultimately

L201: Perhaps the percentage can be deleted or detailed what it refers to e.g. "of the 233 lost SO species"?

L202: "contrary to what modern ecological models would predict (e.g. 3,5)" Do you mean that they would predict southwards range tracking and expansion under global cooling? Perhaps explicitly write out their prediction

L209: The following section, "Threshold response to temperature change", still needs less over-interpretation and greater clarification to the points that your data clearly support, which are sufficient in their own right

L214: "in the order"

L220: I would delete the following as the argument has not been given yet and the sentence does not need it: "once temperature thresholds were surpassed".

L221: Delete "for convenience". We don't make a scientific hypothesis for convenience

L224: Delete the unnecessary: "This magnitude and rate of change is apparently within polar radiolarians' tolerance threshold. "

L231: Perhaps say dropped by an additional 4 degrees, or put 'pre-threshold' in quotation marks as this is just a hypothesis and you don't have much evidence in its support. The remaining problem with the temperature threshold idea is that you only analyse the long-term change, while finer resolution oscillations in temperature, beneath the observational resolution, may be much higher. Furthermore, habitat availability, food supply, changes in currents could all be alternative extinction drivers. The paper is interesting enough without insistence on the temperature threshold idea.

L243: "which is approximately the same magnitude of SST decrease incurred at high latitudes over the last 10 Ma" Except for it is an increase. Polar habitat area should disappear given such an increase, while polar habitat area expands with a global temperature decrease. These are extremely different scenarios, especially given the huge differences in rate of change, and should be acknowledged as such up front. Your results are still perfectly useful, talking about temperature change magnitude. A sentence or two discussing why polar habitat area increase under cooling did not dampen extinctions (or perhaps it did?) might be useful, especially because the isothermal submergence idea suggests all habitats are linked so it should be easy for the organisms to profit from geometrically expanded area at lower latitudes

On fig 2, what about the warming blip in your polar SST time series that is followed (after the sedimentary hiatus) by elevated extinction rates? Might need a mention as it could also be analogous to modern conditions, couldn't it?

L256: " could reduce biodiversity and ecosystem functioning"... in the SO. Actually it could increase both, since you mention the latitudinal cline in diversity increases to the tropics and warmer-adapted immigrant organisms often perform processes faster than cold adapted ones. I think you mean \global\ diversity will likely decline as the SO species are lost. The unknowns with ecosystem functioning may mean it is better to just say functioning is "threatened", or say "if processes are not successfully taken over by immigrated species"

L258: "would likely". Would possibly. See previous comment above

L260: "The loss of these species would also reduce abundances of one of the most common clades". Again, not necessarily. If warm adapted immigrants take over these are often smaller in body size and more numerous, in many taxa. Richness and abundance are not linearly correlated. You suggest range expansions of warm-adapted species in this same paragraph. Probably just delete the reduced abundances part and change to "Such changes could impact both primary producer communities and higher trophic groups..."

L270: "temperature thresholds". Too vague. Be specific: thresholds of temperature change magnitude is the basis for your argument. The paper should be streamlined to make this point clear "with profound consequences for species richness and ecosystem functioning at high latitudes. " Not supported. See above comments and delete

L280: Change "combat" to "replace species lost through" since origination does not diminish extinction

L281: This statement is fine and your argument supports it but in other parts of the MS you need to be honest about what your results do and do not suggest

Fig 2 caption: L321: delete "standard" in standard error bars. L326: Maybe add 'boundary crosser' to 'extinction rate'

L562: "which has been demonstrated to be the most reliable method for paleontological occurrence data". Perhaps tone this down a bit to something like a high-performing method, since this depends on the data and other methods like Pyrate are showing much potential

L563: Statements like this and some of the ones below on generalized least squares (and details) should be in the Results as they help build the storyline: "The mean extinction rate <5Ma was 0.125 (not including youngest edge bin), whereas the mean extinction rate of earlier Neogene time bins (22-5 Ma) was 0.032. "

L569: "Significance of that variable was determined". Isn't it better/clearer to say the significance of differences between timespans was determined?

Reviewer #1 (Remarks to the Author):

The authors satisfactorily addressed all the issues raised in my review. My recommendation is ACCEPT FOR PUBLICATION.

Reviewer #2 (Remarks to the Author):

After reading the comments by the other two reviewers, the arguments put forward by the authors in their rebuttal letter, and the changes they adopted to address the concerns raised, I am happy with the revised manuscript.

The main issues were around the alternative potential extirpation locations to mid latitudes, the ability of the new EEP record to capture low-latitude conditions, the sparseness of data mainly linked to the range-through technique (which the authors replaced with a more viable one, as also suggested by reviewer #3, thus resulting in doubling their sample number), the unclarity of the time and magnitude of the extinction event (now better documented and illustrated, and referred to as an interval), a more rigorous treatment of the discussed trends, and further explanation on the climatic threshold triggering the response. All these, along with a few other minor concerns, have been successfully addressed by the authors.

Reviewer #3 (Remarks to the Author):

I congratulate the authors on a substantially improved paper and a much more convincing storyline. Being perhaps representative of a non-planktonic ecologist reader, I am now convinced about the validity of the observed extinctions, and extinction interval, taking place in the past 5 million years based on the data, analysis and logic presented. I do have some relatively minor remaining issues and addressing these should in no way diminish the importance of the work, but should add to its clarity and thus its utility:

Comment: L16. 'Permanently' is unnecessary. Extinction is always permanent

Response: "Permanently" removed.

Comment: L18. Should 'predicted' be 'assumed' here?

Response: Changed to "assumed."

Comment: The species undergo restructuring? ...or the assemblage undergoes restructuring?

Response: Changed "they" to "assemblages."

Comment: "data from a single section can in principle give a reasonable approximation of radiolarian species occurrences and richness over time throughout a given biome. " In theory, this could be quantitatively supported by data: Take a random well-sampled single core from a well-sampled latitudinal province and see how well (on average) it demonstrates the province species inventory?"

Response: Yes, in theory this could be quantitatively supported by data. However, as our study is the first full species inventory for this province, we are not able to quantitatively test its representativeness.

Comment: L80. Not 'determine'. Some more cautious word would be wise. E.g. "Estimate"

Response: Replaced "determine" with "estimate how many of."

Comment: L93. "has recorded (almost) all species preserved" Not clear how this relates to previous sentence about fossil tropical species richness. All species preserved means what? In the world? In a sample? Do most studies not identify all species in a single sample?

Response: The previous sentence states that tropical species richness has not been fully surveyed, but our study aims to document all species from this province that preserve in the fossil record. Because most studies are focused on biostratigraphy or abundant taxa, they essentially never (at least in published observations) identify all species in a sample.

Comment: L132. Add 'tropical' paleocommunities

Response: Added "tropical."

Comment: L147 and 154: Confusing way to report these statistics e.g, "linear regression two-sided t-test p-values = 0.92 and 0.36 on 9 degrees of freedom for raw and extrapolated species richness trends from 8.5-5Ma, respectively). ". Perhaps say two-sided t-test of linear regression coefficients? Nine degrees of freedom is very few. Not clear why there is so great a difference vs line 154? Would be nice to see the coefficients/means being compared too, also for line 154

Response: Language reworded to be more clear. Changed to: "probability of no change in species richness versus time during 8.5-5Ma: 0.92 [raw], and 0.36 [extrapolated], both two-tailed t-tests with 9 degrees of freedom." And: "probability of no change in species richness versus time during 5-0Ma: $8.55e-10$ [raw], and $3.30e-5$ [extrapolated], both two-tailed t-tests with 42 degrees of freedom."

Comment: L152: "Many of the species marginalized by the rise in *Antarctissa* " Can we really say this? Perhaps *Antarctissa* just made use of niche space already vacated by the other species, or their niches did not overlap at all and the rise and falls are simple correlations responding to some other driver?

Response: Changed to "Many of the species that became rare as *Antarctissa* became common [...]"

Comment: L162. 71% of the species losses were...: 'Was' to 'were', or just 'went extinct'. So that the reader understands your logic, perhaps add the justification here to, "71% of of the 233 polar species lost were not observed in coeval or younger low-latitude sediments, so we infer that they went extinct".

Response: Changed to "went extinct." We explain justification for this in the following sentence, so it would be repetitive to add here too.

Comment: L183: "233 SO species went extinct" I thought 71% of this number went extinct? Or

is this number referring to a different thing than the previous '233' (L162)?

L193: "the SO species impacted by high-latitude cooling": Is this of the 233 species lost again? Or simply all of the species that were polar at the time?

Response: Thank you for catching that typo. 233 species were lost from the SO, 71% of which were lost due to extinction (166 species). In line 193 (line 252 in revised manuscript) the "species impacted" refers to the number of species lost (233), which includes those that went extinct along with those that were extirpated.

Comment: L199: "as they submerged ultimately died". 'And' ultimately

Response: Typo fixed.

Comment: L201. Perhaps the percentage can be deleted or detailed what it refers to e.g. "of the 233 lost SO species"?

Response: Added "out of 233."

Comment: L202: "contrary to what modern ecological models would predict (e.g. 3,5)" Do you mean that they would predict southwards range tracking and expansion under global cooling? Perhaps explicitly write out their prediction

Response: Yes, they predict that species will shift toward higher latitudes. Sentence added: "Most models parameterize biogeographic response primarily by changing temperature, thus predicting gradual northward shifts toward the tropics in response to global cooling for Southern Ocean species."

Comment: L209: The following section, "Threshold response to temperature change", still needs less over-interpretation and greater clarification to the points that your data clearly support, which are sufficient in their own right

Response: We have clarified in L262-266 that because in the modern ocean temperature is the most significant environmental predictor of radiolarian and other plankton species abundances and distributions, most models parameterize biogeographic response primarily by changing temperature, and thus would predict gradual northward shifts towards the tropics in response to cooling for SO species. However, our data do not support these models of a gradual biogeographic shift, so we instead interpret the results as a threshold response to temperature change.

Comment: L214: "in the order"

Response: Changed to "of approximately"

Comment: L220: I would delete the following as the argument has not been given yet and the sentence does not need it: "once temperature thresholds were surpassed".

Response: We mentioned that the temperature threshold was not surpassed for tropical radiolarians, so for clarity we choose to use the same language for discussing the SO radiolarian response.

Comment: L221: Delete “for convenience”. We don’t make a scientific hypothesis for convenience

Response: “Convenience” removed.

Comment: L224: Delete the unnecessary: “This magnitude and rate of change is apparently within polar radiolarians’ tolerance threshold.”

Response: Sentence deleted.

Comment: L231: Perhaps say dropped by an additional 4 degrees, or put ‘pre-threshold’ in quotation marks as this is just a hypothesis and you don’t have much evidence in its support. The remaining problem with the temperature threshold idea is that you only analyse the long-term change, while finer resolution oscillations in temperature, beneath the observational resolution, may be much higher. Furthermore, habitat availability, food supply, changes in currents could all be alternative extinction drivers. The paper is interesting enough without insistence on the temperature threshold idea.

Response: Changed “pre-threshold” to “Phase 1.”

Comment: L243: “which is approximately the same magnitude of SST decrease incurred at high latitudes over the last 10 Ma” Except for it is an increase. Polar habitat area should disappear given such an increase, while polar habitat area expands with a global temperature decrease. These are extremely different scenarios, especially given the huge differences in rate of change, and should be acknowledged as such up front. Your results are still perfectly useful, talking about temperature change magnitude. A sentence or two discussing why polar habitat area increase under cooling did not dampen extinctions (or perhaps it did?) might be useful, especially because the isothermal submergence idea suggests all habitats are linked so it should be easy for the organisms to profit from geometrically expanded area at lower latitudes. On fig 2, what about the warming blip in your polar SST time series that is followed (after the sedimentary hiatus) by elevated extinction rates? Might need a mention as it could also be analogous to modern conditions, couldn’t it?

Response: In line 320-322 we explain that the difference in the direction of ongoing temperature change (warming) could mean more severe extinction in SO assemblages than observed in our study. Also, we do acknowledge the rate differences between the past and future scenarios in the previous sentence. We agree it would be interesting to investigate the “warming blip” in polar SST, but without radiolarian data from the hiatus interval directly preceding it, we do not wish to speculate.

Comment: L256: “ could reduce biodiversity and ecosystem functioning”... in the SO. Actually it could increase both, since you mention the latitudinal cline in diversity increases to the tropics and warmer-adapted immigrant organisms often perform processes faster than cold adapted ones. I think you mean \global\ diversity will likely decline as the SO species are lost. The unknowns with ecosystem functioning may mean it is better to just say functioning is “threatened”, or say “if processes are not successfully taken over by immigrated species”

Response: Changed “reduce” to “threaten.”

Comment: L258: “would likely”. Would possibly. See previous comment above

Response: Changed “likely” to “possibly.”

Comment: L260: “The loss of these species would also reduce abundances of one of the most common clades”. Again, not necessarily. If warm adapted immigrants take over these are often smaller in body size and more numerous, in many taxa. Richness and abundance are not linearly correlated. You suggest range expansions of warm-adapted species in this same paragraph. Probably just delete the reduced abundances part and change to “Such changes could impact both primary producer communities and higher trophic groups...”

Response: Phrase removed.

Comment: L270: “temperature thresholds”. Too vague. Be specific: thresholds of temperature change magnitude is the basis for your argument. The paper should be streamlined to make this point clear

Response: Changed to specify that the temperature threshold we recognized was ~6°C for radiolarians.

Comment: “with profound consequences for species richness and ecosystem functioning at high latitudes. ” Not supported. See above comments and delete

Response: Changed “profound” to “important”. We have rebutted the reviewer’s criticisms regarding our temperature response interpretation, and thus feel that this statement is supported by our data. We are willing, however, to temper our observation by using a less dramatic descriptive word.

Comment: L280: Change “combat” to “replace species lost through” since origination does not diminish extinction

Response: Changed “combat” to “offset.”

Comment: L281: This statement is fine and your argument supports it but in other parts of the MS you need to be honest about what your results do and do not suggest

Response: We have made our best attempt to do this.

Comment: Fig 2 caption: L321: delete “standard” in standard error bars.

Response: The editor requested that we describe the type of error bars, so we kept “standard” in the caption.

Comment: L326: Maybe add ‘boundary crosser’ to ‘extinction rate’

Response: “Boundary crosser” added.

Comment: L562: “which has been demonstrated to be the most reliable method for paleontological occurrence data”. Perhaps tone this down a bit to something like a high-performing method, since this depends on the data and other methods like Pyrate are showing much potential

Response: Changed “the most” to “a.”

Comment: L563: Statements like this and some of the ones below on generalized least squares (and details) should be in the Results as they help build the storyline: “The mean extinction rate <5Ma was 0.125 (not including youngest edge bin), whereas the mean extinction rate of earlier Neogene time bins (22-5 Ma) was 0.032. ”

Response: This result was added to the main text (L296-298).

Comment: L569: “Significance of that variable was determined”. Isn’t it better/clearer to say the significance of differences between timespans was determined?

Response: Changed to “significance of differences between timespans.”